# Astrocytic-OTUD7B ameliorates murine experimental autoimmune encephalomyelitis by stabilizing glial fibrillary acidic protein and preventing inflammation

Kunjan Harit[1], Wenjing Yi[1], Andreas Jeron[2], Jakob Schmidt[3], Ruth Beckervordersandforth [3], Emanuel Wyler [4], Artür Manukyan[4], Martina Deckert[5], Helena Radbruch [6], Thomas Conrad [7], Janine Altmüller[7,8], Markus Landthaler [4,9], Xu Wang [1,10,12], Gopala Nishanth [1,12] & Dirk Schlüter [1,11,12] ✉

Astrocytes are central to the pathogenesis of multiple sclerosis (MS); however, their regulation by post-translational ubiquitination and deubiquitination is unresolved. This study shows that the deubiquitinating enzyme OTUD7B in astrocytes protects against murine experimental autoimmune encephalomyelitis (EAE), a model of MS, by limiting neuroinflammation. RNA-sequencing of isolated astrocytes and spatial transcriptomics show that in EAE, OTUD7B downregulates chemokine expression in astrocytes of inflammatory lesions, which is associated with reduced recruitment of encephalitogenic CD4$^+$ T cells. Furthermore, OTUD7B is necessary for glial fibrillary acidic protein (GFAP) expression of astrocytes bordering inflammatory lesions. Mechanistically, OTUD7B (i) restricts TNF-induced chemokine production of astrocytes by sequential K63- and K48-deubiquitination of RIPK1, which limits NF-κB and MAPK activation and (ii) enables GFAP protein expression by supporting GFAP mRNA expression and preventing its proteasomal degradation through K48-deubiquitination of GFAP. This dual action on TNF signaling and GFAP identifies OTUD7B as a central inhibitor of astrocyte-mediated inflammation.

Astrocytes are a highly abundant cell population present in all regions of the central nervous system (CNS) and play an important role in the maintenance of CNS homeostasis and health. Under physiological conditions, astrocytes provide physical and metabolic support for neurons and blood-brain barrier (BBB) functions[1–3]. Upon CNS damage, astrocytes are rapidly activated and undergo structural and functional changes, by a process known as reactive astrogliosis which critically regulates the pathogenesis, development and outcome of CNS

disorders including autoimmune diseases, infections, neurodegenerative diseases and trauma[4]. Reactive astrogliosis is characterized by astrocyte hypertrophy and increased STAT3-mediated expression of the glia fibrillary acidic protein (GFAP)[5].

Depending on the underlying disease and the inflammatory milieu astrocytes can either support or suppress CNS inflammation, contribute to CNS damage but also promote regeneration. This functional plasticity of astrocytes is based on the activation of different signaling

pathways leading to the respective production and secretion of disease-modifying proteins. In multiple sclerosis (MS), a human inflammatory demyelinating disease and its murine model experimental autoimmune encephalitis (EAE), reactive astrocytes upregulate GFAP protein expression and are the major producers of chemokines[6,7]. These chemokines foster the recruitment of immune cells, primarily autoimmune CD4[+] T cells to the site of inflammation. The CNS infiltrating CD4[+] T cells orchestrate an attack on the myelin sheath resulting in demyelination and neurodegeneration[8,9]. The production of chemokines by astrocytes in MS and EAE is driven by several signaling pathways, including NF-κB, MAPK and JAK-STAT pathways[10–12]. Experimental studies have also shown that astrocyte-specific ablation of NF-κB signaling molecules attenuates EAE[11,12]. In addition to disease promoting functions, astrocytes can exert protective roles, in particular, at later stages of EAE and MS. These protective mechanisms include the local restriction and resolution of neuroinflammation as well as the promotion of remyelination and axonal repair[13–15]. In this regard, STAT3-mediated astrocyte survival, proliferation and up-regulation of GFAP expression ameliorates EAE by limiting the spread of encephalitogenic T cells and bordering of inflammatory lesions[16]. Depending on the subtype of MS and the EAE model, active and resolving inflammatory lesions exist in parallel throughout the CNS. In these different lesions astrocyte reactivity is diverse, shows a high plasticity, differs regionally and may also change over time[17–19].

Astrocyte reactivity is significantly determined by the differential activation of signaling pathways which regulate inflammatory reactions and glia scar formation. These signaling pathways are critically modulated by ubiquitination, a posttranslational modification mediated by the covalent linkage of ubiquitin, a 76-aa large protein, to substrates. Ubiquitination is catalyzed sequentially by an ubiquitin-activating enzyme (E1), an ubiquitin-conjugating enzyme (E2), and an ubiquitin ligase (E3). Ubiquitin molecules can be added to proteins in the form of monomers and polyubiquitin chains, in which ubiquitins are linked through the N-terminal methionine residue (M1) or one of the seven lysine residues (K6, K11, K27, K29, K33, K48, K63)[20]. The fate of ubiquitinated substrates is determined by the type of ubiquitin linkage. K48- and K11 chains lead to proteasomal degradation of substrates, whereas K63-linked ubiquitin chains modify protein function and can trigger signal transduction[21–24]. The process of ubiquitination is reversible and can be counteracted by deubiquitinating enzymes (DUBs). As fine-tuning modulators of cell signaling and activities, DUBs have emerged as important regulators of astrocytes and CNS auto-immunity mediating both protective and disease-promoting astrocyte functions[25]. Of note, the function of DUBs in CNS autoimmunity is also cell type-specific as illustrated by EAE promoting function of the DUB A20 (TNFAIP3) in T cells and its EAE inhibitory function in astrocytes[26,27].

OTUD7B is a DUB belonging to the OTU subfamily and is expressed in all human and murine tissues. OTUD7B is expressed in all regions of the CNS, in particular in astrocytes and oligodendrocytes but not in neurons and microglia (https://www.proteinatlas.org/ENSG00000264522-OTUD7B). OTUD7B can hydrolyze K11-[28], K48-[29] and K63-[30] linked ubiquitin chains from distinct substrates and regulates pro-inflammatory signaling by inhibiting TNF-mediated NF-κB activation through K63 deubiquitination of RIPK1 and TRAF6[30,31]. The regulation of TNF signaling by OTUD7B is cell type-specific since OTUD7B prevents TNF-mediated apoptosis in dendritic cells in vitro and in murine cerebral malaria by cleaving K48-ubiquitin chains from the E3 ubiquitin ligase TRAF2[29]. In T cells, OTUD7B inhibits TCR and CD28-mediated activation by regulating the non-degradative ubiquitination of ZAP70, a central molecule in proximal TCR signaling[32]. Consequently, OTUD7B-deficient mice are protected from myelin oligodendrocyte glycoprotein (MOG)-induced EAE due to impaired activation of encephalitogenic CD4[+] T cells[32] further illustrating cell

type- and disease-specific functions of OTUD7B in inflammatory diseases.

In this study we investigated the unresolved astrocyte-specific function of OTUD7B in EAE and identified a dual role of OTUD7B. First, OTUD7B restricts TNF-induced pro-inflammatory astrocyte responses by sequential K63- and K48-deubiquitination of RIPK1. Second, OTUD7B fosters GFAP protein expression and astrogliosis by enhancing interleukin (IL) − 6 STAT3-dependent GFAP mRNA expression and preventing proteasomal degradation of GFAP through its K48-deubiquitination.

## Results

### Upregulation of astrocytic Otud7b during CNS autoimmunity
To investigate the role of OTUD7B in CNS autoimmunity, we first compared the expression of *Otud7b* in the astrocytes of patients with progressive MS and healthy individuals. Analysis of public available microarray dataset (No. GSE180759) revealed that *Otud7b* mRNA expression was upregulated in the lesion edges and periplaque white matter of MS patients as compared to control white matter of healthy individuals (Fig.1A). Next, we determined the expression of *Otud7b* in MOG-induced EAE, which is characterized by a peak of disease around day 15 post immunization (p.i.) followed by a gradual decline of clinical symptoms[16]. In EAE, *Otud7b* mRNA expression was upregulated and highest in spinal cords at day 15 p.i. and declined up to day 22 p.i. (Fig. 1B). Immunofluorescence imaging of the spinal cord further demonstrated a marked increase of OTUD7b protein expression in the astrocytes at 15 p.i. as compared to non-immunized mice (Fig. 1C).

Consistently, analysis of *Otud7b* mRNA expression of astrocytes isolated from the normal and EAE-diseased spinal cord revealed a prominent upregulation of *Otud7b* mRNA expression in astrocytes of mice with EAE (Fig. 1D). For a detailed spatial analysis of the expression of astrocytic *Otud7b* in EAE, we performed spatial transcriptomics with single-molecule resolution on the Xenium platform. Upon immunization with MOG$_{35-55}$ peptide and a pertussis toxin boost, cell types were identified based on the transcriptome data using the pre-designed mouse brain panel with 247 genes in combination with a custom panel including additional 50 genes (Suppl. dataset. 1). We defined inflammatory lesions in the sections characterized by the accumulation of CD3[+] CD4[+] and CD3[+] CD8[+] T cells (Fig. 1E). In addition, we defined a lesion rim in direct proximity to the lesions, with fewer T cells (200 μm), and as a third region defined peri-lesion with no T cells (Fig. 1E, black lines are defined lesion regions). We identified that *Otud7b* mRNA is upregulated about 1.5-fold in Sox2[+] Sox9[+] Aqp4[+] astrocytes within the lesion core and lesion rim, but not in the peri-lesions as compared to non-immunized mice (Fig. 1F). As expected, the number of identified cells per section increased almost two-fold upon immunization (Fig. 1G), with a strong influx of CD68[+] microglia and macrophages into the lesions (Fig. 1G). We observed a strong induction of genes associated with inflammation including Isg15, the chemokines CXCL9, Ccl2, CXCL10 and the cytokines IL-6 and IL1β at d15 p.i. (Fig. 1H), with the induction being particularly strong within the lesion core and a gradual decline in the lesion rim and peri-lesion. Taken together, astrocytes upregulate *Otud7b* expression in CNS auto-immunity with the highest expression in the T cell-enriched lesions and a gradual decline with increasing distance from the lesions core.

### Astrocytic OTUD7B-deficiency aggravates EAE
To determine the astrocyte-specific function of OTUD7B in EAE, we crossed GFAP-cre mice with Otud7b[fl/fl] to generate GFAP-cre Otud7b[fl/fl] mice with deletion of *Otud7b* in astrocytes. The GFAP-cre strain used here expresses Cre late during embryonic development (day 14.5[33]), has a high deletion efficacy in all spinal cord and brain astrocytes but only low deletion in neurons[10,11,16,33], and allows analysis of astrocyte functions in inflammatory CNS disorders[10,11,16,17,34]. Mice were genotyped using DNA extracted from ear biopsies. Knockout mice were

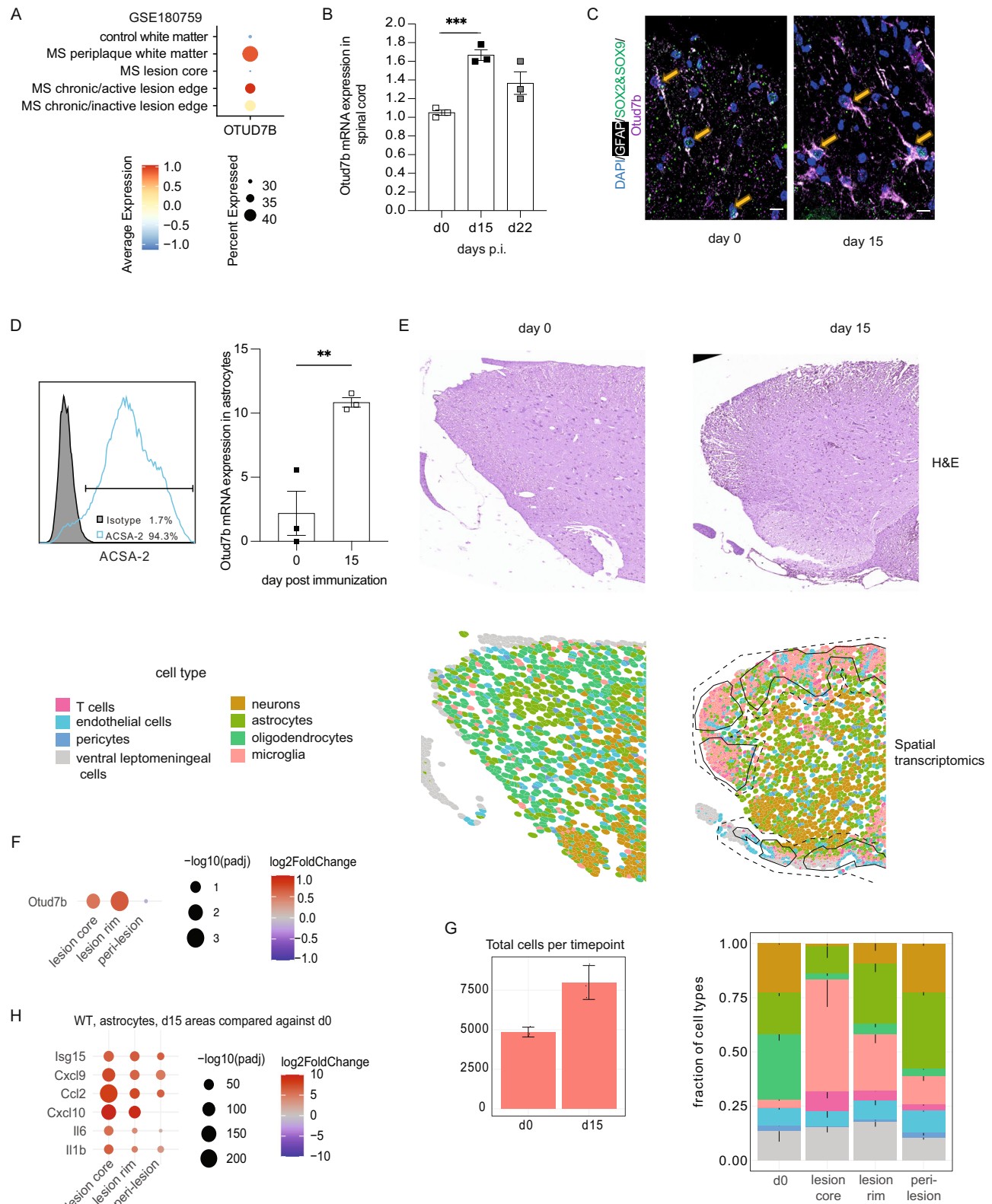

confirmed by the presence of the floxed OTUD7B allele and Cre recombinase under the GFAP promoter, while wild-type controls carried only the floxed OTUD7B allele (Suppl. Fig. 1A).In vivo deletion of *Otud7b* in spinal cord astrocytes was validated by spatial transcriptomics showing a strong upregulation of *Otud7b* in astrocytes in EAE of Otud7b^fl/fl mice, which was absent in GFAP-cre Otud7b^fl/fl mice (Fig. 2A). In Otud7b^fl/fl mice, *Otud7b* was most strongly expressed in astrocytes, much weaker in neurons and marginally in pericytes,

microglia, oligodendrocytes and endothelial cells (Fig.2A). Western blot (WB) analysis of astrocytes isolated from spinal cord of Otud7b^fl/fl and GFAP-Cre Otud7b^fl/fl also demonstrated efficient deletion of OTUD7B in astrocytes of GFAP-cre Otud7b^fl/fl mice (Supplementary Fig. 1B). GFAP-cre Otud7b^fl/fl mice were born in a normal mendelian ratio and grew normally. Histological analysis showed that astrocytes in the spinal cords of GFAP-cre Otud7b^fl/fl mice lacked OTUD7B expression (Supplementary Fig. 1C). Despite this, the spinal cords

**Fig. 1 | Upregulation of Otud7b in MS and EAE lesions. A** mRNA levels of OTUD7b were analyzed from astrocytes from different lesional regions of patients with progressive MS (n = 5), and control white matter of astrocytes from healthy individuals (n = 3). Data from GEO database (accession number GSE180759). **B** EAE was induced in Otud7b^fl/fl mice by MOG$_{35-55}$ peptide immunization combined with pertussis toxin. Spinal cord was isolated from non-immunized (d0) and immunized mice at day15 and day 22 days p.i., mRNA was isolated and the expression of *Otud7b* mRNA was analyzed by qRT-PCR. (n = 3). **C** Immunofluorescence staining showing presence of SOX2/SOX9+GFAP+ OTUD7b expressing astrocytes (marked by yellow arrow) in spinal cord sections of unimmunized and day 15 p.i. Otud7b^fl/fl mice. All photographs are representative of three mice per group. Scale bars correspond to 10 μm. **D** Astrocytes were isolated from non-immunized (d0) and immunized mice at day15 p.i., using anti-ACSA-2 antibody for magnetic cell sorting. The expression of *Otud7b* mRNA in sorted astrocytes was analyzed by qRT-PCR (n = 3 mice per timepoint). (E-H) For spatial transcriptomics, EAE was induced in Otud7b^fl/fl mice by MOG$_{35-55}$ peptide immunization combined with pertussis toxin. 5 μm thick sections of the spinal cord from non-immunized and immunized mice at day15 (n = 3, both genotypes) were placed on Xenium slides and in situ RNA expression analysis was performed using a pre-designed mouse brain panel of 247 genes and a custom panel of 50 genes. (E, top panel) Representative hematoxylin and eosin (H&E)-stained sections from Otud7b^fl/fl mice (d0 and d15 p.i.) mounted on Xenium slides. **E** (bottom panel) Cell map from the same mice showing the distribution of the annotated cell populations in the spinal cord based on single-cell sequencing analysis using Xenium mouse brain slides in their native position. In mice with EAE, the inner area surrounded by black solid lines represents the lesion core compartment, he area between the black solid and dashed lines represents the lesion rim compartment and the outermost peri-lesion compartment is on the other site of the dashed line. **F** Dot plot showing *Otud7b* expression at day 15 p.i. in different lesion compartments compared to non-immunized mice. **G** The left bar graph shows the total number of cells in the spinal cord sections and the right stacked bar graph shows the relative cellular composition in different anatomical compartments (thin vertical lines indicate standard error across three replicates for the percentage of the respective cell type). (n = 3 mice per timepoint). **H** Chemokine gene expression by astrocytes at day15 p.i. compared to astrocytes of non-immunized mice in the different disease-associated anatomical compartments. All data represented as mean values ± SEM. *p < 0.05, **p < 0.01, ***p < 0.001, ****p < 0.0001. ns, not significant. Statistical analyses: two-tailed Student's t test (**B, D**) Adjusted p-values were calculated by DEseq2 using Benjamini-Hochberg corrections of two-sided Wald test p values (**F, G, H**). Scale bar = 10 μm (**C**),(**E**) 200 μm. Source data are provided as source data file.

---

appeared normal, with no signs of neurodegeneration or inflammation (Supplementary Fig. 1D). Major leukocyte populations in the spinal cord, spleen, and lymph node were comparable between Otud7b^fl/fl and GFAP-cre Otud7b^fl/fl mice (Supplementary Fig. 1E–H), consolidating that OTUD7B expression in astrocytes does not regulate inflammatory responses under homeostatic conditions.

To explore the role of astrocytic OTUD7B in EAE, we immunized Otud7b^fl/fl and GFAP-cre Otud7b^fl/fl mice with MOG peptide. Although the two genotypes had similar disease onset at around day 12 p.i., GFAP-cre Otud7b^fl/fl mice showed significantly increased clinical scores with higher maximal clinical scores and disease incidence (Fig. 2B), and reduced body weight (Fig. 2C).

In accordance with the aggravated clinical symptoms, and reduced myelin thickness was more pronounced in GFAP-cre Otud7b^fl/fl as compared to control mice at d15 p.i. (Fig. 2D). In addition, inflammatory infiltrates were larger, more confluent and infiltrated deeper in the spinal cord tissue in GFAP-cre Otud7b^fl/fl mice (Fig. 2D, E). In EAE, astrocyte morphology and reactivity was strongly altered. Adjacent to and bordering inflammatory lesions astrocytes of Otud7b^fl/fl mice strongly expressed GFAP and showed prolonged GFAP+ astrocytic processes (Fig. 2F, G). On the contrary, astrocytes of the GFAP-cre Otud7b^fl/fl mice associated with inflammatory infiltrates only weakly expressed GFAP and, thus, did not form a GFAP+ border surrounding the inflammatory infiltrate (Fig. 2F–H). In both control and GFAP-cre Otud7b^fl/fl mice, equal numbers of SOX2+ SOX9+ astrocytes were present indicating that the reduction of GFAP expression was not caused by a loss of OTUD7B-deficient astrocytes (Fig. 2G–H). Interestingly, a comparison of GFAP mRNA levels by spatial transcriptomics showed upregulation particularly in the lesion core and the lesion rim, which was higher in Otud7b^fl/fl than in GFAP-cre Otud7b^fl/fl mice. (Fig. 2I) indicating that the diminished transcription of GFAP may contribute to the reduced GFAP protein in GFAP-cre Otud7b^fl/fl mice (Fig.2F–H).

## OTUD7B suppressed chemokine and cytokine production by astrocytes

Reactive astrocytes contribute to the development of EAE by producing pro-inflammatory mediators, including chemokines, which induce the recruitment of encephalitogenic autoimmune CD4+ T cells into the CNS[35,36]. Since GFAP-cre Otud7b^fl/fl mice exhibited increased CNS inflammation, we next determined the impact of OTUD7B on the transcriptome of astrocytes isolated from the spinal cords of non-immunized and MOG-immunized (d15 p.i.) Otud7b^fl/fl and GFAP-cre Otud7b^fl/fl mice. We detected 1944 genes differentially regulated between OTUD7B-deficient and -sufficient astrocytes (Fig. 3A). Of these 1944 genes, 689 genes were differentially regulated under homeostatic conditions, 1090 genes were regulated at d15 p.i. and 165 genes were regulated under both conditions (Fig. 3A). Clustering of the genes using the K-means clustering algorithm resulted in 10 clusters. Ontology and KEGG pathway enrichment analysis showed that the pathways related to chemokine and cytokine signaling (Fig. 3B) were upregulated in cluster 6 and 2 of OTUD7B-deficient astrocytes upon EAE. In addition, OTUD7B-deficient astrocytes of cluster 6 and 2 had increased expression of genes regulating transendothelial migration of leukocytes, pro-inflammatory cytokine and chemokine signaling, cell adhesion and angiogenesis (Fig. 3C–D). A detailed analysis of chemokine genes showed that upon induction of EAE, expression of CCL and CXCL chemokines was upregulated in both genotypes but expression of CCL2, 3, 4, 5, 6, 7, 8, 11, 19 and CXCL 1, 2, 9, 10, 12, 13, 16 were higher in OTUD7B-deficient astrocytes. This suggests that the increased disease pathology in GFAP-cre Otud7b^fl/fl mice was due to increased chemokine and cytokine mediated infiltration of leukocytes into the spinal cord, (Fig. 3E). Only Ccl22, Cxcl16, and Cxcl12, which are stronger expressed by microglia and brain macrophages as compared to astrocytes (https://www.proteinatlas.org) were increased upregulated in OTUD7B-comptetent astrocytes (Fig. 3E).

Spatial transcriptome analysis of spinal cord tissue provided further information on the anatomic distribution of chemokines and cytokines. Chemokine mRNA expression was higher in OTUD7B-deficient astrocytes in the lesion cores and in the rims surrounding the lesion in relation to peri-lesion areas where the chemokine levels were comparable (Fig. 3F). Among the cytokines studied, only IL-6 and IL-1β but not TNF, TGF-β1 were highly expressed in OTUD7B-deficient astrocytes in EAE. In contrast to astrocytes, chemokine and cytokine expression of microglia was equal between the two genotypes.

Consistent with the astrocyte RNA-sequencing data, RT-PCR-based quantification of chemokine mRNA in spinal cords of mice with EAE revealed that GFAP-cre Otud7b^fl/fl mice had significantly higher levels of CXCL1, CXCL10, CXCL11, CCL2, and CCL20 mRNA at day 15 p.i., which are major attractants for lymphoid and monocytic cells (Fig. 3G).

## Increased recruitment of encephalitogenic CD4+ T cells in GFAP-cre Otud7b^fl/fl mice

To determine whether the increased chemokine production of OTUD7B-deficient astrocytes resulted in an enhanced recruitment of leukocytes to the CNS in MOG-immunized mice, we performed a flow cytometry analysis of leukocytes isolated from the spinal cords. In contrast to non-immunized mice (Supplementary Fig. 1E), relative and

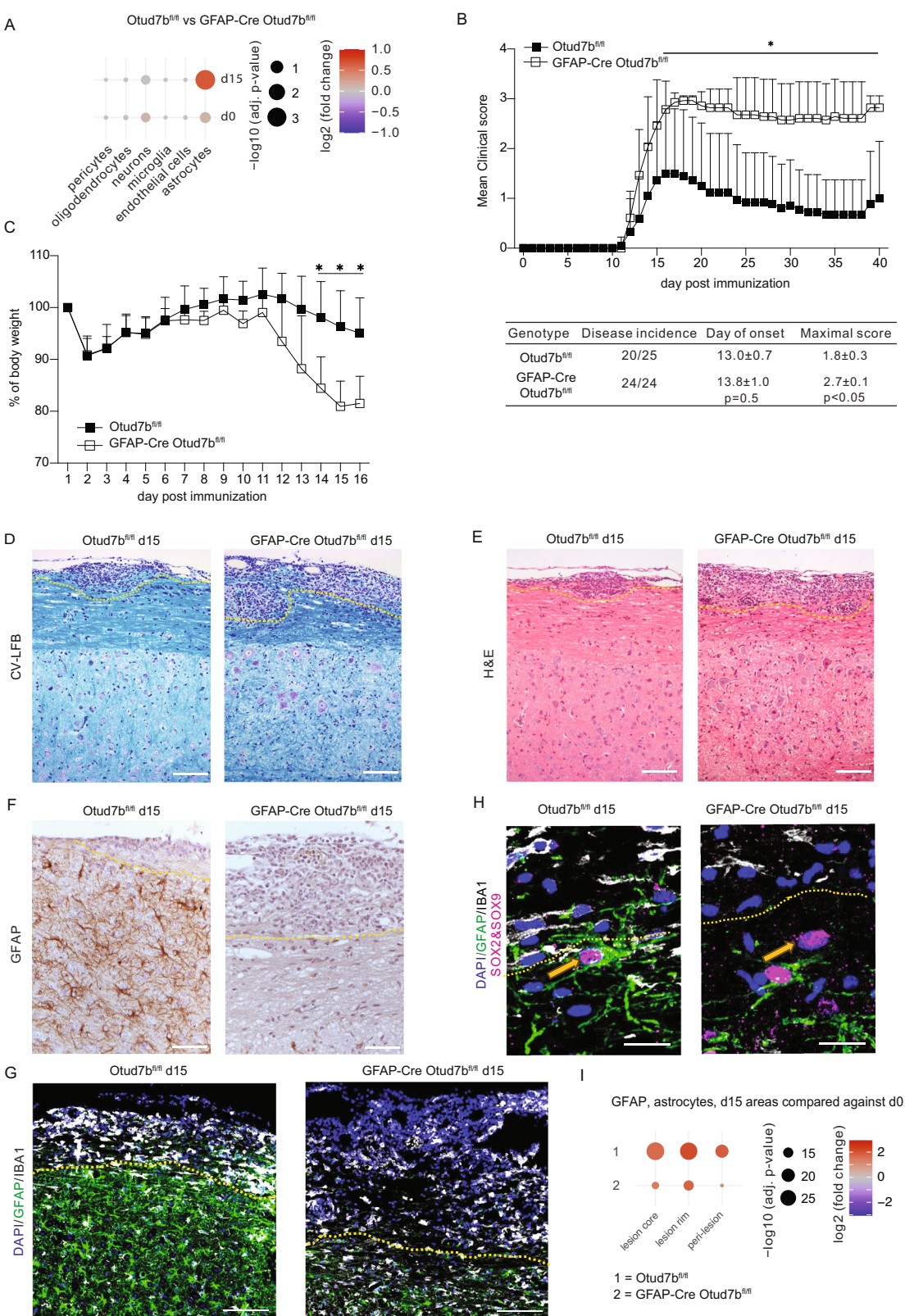

| Genotype | Disease incidence | Day of onset | Maximal score |
|---|---|---|---|
| Otud7b$^{fl/fl}$ | 20/25 | 13.0±0.7 | 1.8±0.3 |
| GFAP-Cre Otud7b$^{fl/fl}$ | 24/24 | 13.8±1.0 p=0.5 | 2.7±0.1 p<0.05 |

absolute numbers of CD4$^+$ but not of CD8$^+$ T cells were significantly increased in Otud7b$^{fl/fl}$ and GFAP-cre Otud7b$^{fl/fl}$ mice with EAE (Fig. 4A). Additionally, numbers of CD11c$^+$ dendritic cells, Ly6C$^{high}$ CD11b$^+$ inflammatory monocytes and F4/80$^+$ CD11b$^+$ macrophages/microglia but not of CD19$^+$ B cells were higher in GFAP-cre OTUD7b$^{fl/fl}$ mice (Fig. 4B). Among CD4$^+$ T cell subsets, IFN-producing Th1 cells, IL-17-producing Th17 cells, and GM-CSF-producing CD4$^+$ T cells are the

major effectors in EAE and each of them can induce EAE independently[37–39]. We identified that the absolute but not relative numbers of infiltrating GM-CSF$^+$, IFN-γ$^+$, and IL17$^+$ CD4 cells were significantly increased in the spinal cord of GFAP-cre Otud7b$^{fl/fl}$ mice with EAE (Fig. 4C, D). This indicates that astrocyte-specific OTUD7B limits the recruitment of encephalitogenic CD4$^+$ T cell subsets to the CNS but does not influence the differentiation and composition of the recruited

**Fig. 2 | *Astrocyte-specific OTUD7B ameliorates EAE.*** EAE was induced in Otud7b[fl/fl] and GFAP-cre Otud7b[fl/fl] by MOG[35-55] peptide immunization combined with pertussis toxin (**A**) Spatial transcriptomics of Otud7b mRNA expression in CNS-resident cell populations of non-immunized and immunized Otud7b[fl/fl] and GFAP-cre Otud7b[fl/fl] mice. **B** The mice were monitored daily for the clinical signs of the disease. Upper panel represents mean clinical score ± SEM (Otud7b[fl/fl] n = 7, GFAP-cre Otud7b[fl/fl] n = 11) and the lower panel shows the disease incidence, day of disease onset and maximum clinical score of all immunized mice per group. Data represented as mean values ± SEM. **C** The body weight was measured daily up to day 16 p.i. Graph represents percent body weight ± SEM (Otud7b[fl/fl] n = 7, GFAP-cre Otud7b[fl/fl] n = 5). All bars represent mean values ± SEM. Student's t-test *p < 0.05, **p < 0.01. **D** Cresyl-violet luxol-fast blue staining showing inflammatory infiltrates (marked by dotted line) and myelin staining at day 15 p.i in spinal cord of Otud7b[fl/fl] and GFAP-cre Otud7b[fl/fl] mice. **E** H&E staining showing immune cell infiltration (marked by dotted line) in spinal cord sections from Otud7b[fl/fl] and GFAP-cre Otud7b[fl/fl] mice at day 15 p.i. **E–G** At day 15 p.i., reactive astrocytes of Otud7b[fl/fl] mice exhibit strong GFAP immunoreactivity close to inflammatory lesions in the spinal cord, whereas GFAP immunoreactivity of astrocytes surrounding inflammatory lesions in spinal cord sections of GFAP-cre Otud7b[fl/fl] mice was largely lost by immunohistochemistry (**F**) and immunofluorescence (**G**) All photographs are representative of three mice per group. Scale bars correspond to 100μm (**D**–**G**). **H** Immunofluorescence staining showing presence of SOX2/SOX9⁺GFAP astrocytes surrounding inflammatory lesions (marked by yellow dotted line) in spinal cord sections of Otud7b[fl/fl] and GFAP-cre Otud7b[fl/fl] mice at day 15 p.i. All photographs are representative of three mice per group. Scale bars correspond to 20μm. **I** Dot plot showing GFAP mRNA expression in different lesion compartments at d 15 p.i. *p < 0.05, **p < 0.01, ***p < 0.001, ****p < 0.0001. ns, not significant. Statistical analyses: Mann–Whitney U test (**B**) two-tailed Student's t test (**C**), Adjusted p-values were calculated by DEseq2 using Benjamini–Hochberg corrections of two-sided Wald test p values (**A**, **I**). No adjustments were made for the multiple comparisons. Source data are provided as source data file.

CD4⁺ T cell subsets during EAE. In accordance with the increased recruitment of encephalitogenic CD4⁺ T cells, inflammatory monocytes and macrophages into the CNS of GFAP-cre Otud7b[fl/fl] mice with EAE, mRNA production of IFN-γ,TNF, IL-17, GM-CSF and NOS2, which all contribute to demyelination in EAE[40–44], was also increased (Fig. 4E).

Collectively, OTUD7B expression in astrocytes suppressed astrocyte chemokine production, recruitment of encephalitogenic leukocytes, demyelination and disease symptoms in EAE illustrating the important neuroprotective function of astrocytic OTUD7B.

## OTUD7B suppressed early TNF-induced pro-inflammatory signaling in astrocytes

The increased astrocyte activation and chemokine production of GFAP-cre Otud7b[fl/fl] mice under inflammatory but not under homeostatic conditions indicates that pro-inflammatory cytokines induced OTUD7B-dependent immunoregulatory astrocyte function. To identify whether astrocyte chemokine and cytokine production induced by the encephalitogenic cytokines TNF, IFN-γ and IL-17, respectively, is regulated by OTUD7B, we cultured OTUD7B-deficient and -competent astrocytes, stimulated them with the cytokines and determined chemokine and cytokine production by RT-PCR. Upon stimulation with TNF, mRNA expression of CXCL1, CXCL11, CCL20, IL-6, CCL2 and NOS2 were significantly increased in OTUD7B-deficient astrocytes (Fig. 5A). In contrast, OTUD7B did not regulate IFN-γ- and IL-17-induced chemokine and cytokine production (Fig. 5A) indicating that TNF signaling is the major pathway regulated by OTUD7B in astrocytes. Thus, both increased TNF mRNA expression in EAE (Fig. 4E) and TNF-induced astrocytic cytokine and chemokine production are under control of astrocytic OTUD7B.

To directly assess the effect of OTUD7B on TNF, IFN-γ and IL-17 signaling, we stimulated cultivated astrocytes with these cytokines and analyzed activation of the respective signaling pathways by WB. Upon stimulation with TNF, activation of both the NF-κB and MAPK pathway, indicated by increased phosphorylation of p65 and degradation of IκBα, and the phosphorylation of ERK, p38 and JNK in the MAPK pathways were stronger in OTUD7B-deficient astrocytes (Fig. 5B). Of note, these differences were detectable as early as 10 min post TNF stimulation. In contrast, stimulation with IFN-γ did not result in increased STAT1 phosphorylation (Fig. 5C) and activation with IL-17 had no effect on phosphorylation of IκBα, ERK, p38 and JNK (Fig. 5D). Thus, OTUD7B is an inhibitor of TNF but not IFN-γ and IL-17 signaling in astrocytes. To further prove that OTUD7B inhibits TNF-induced inflammation in astrocytes, OTUD7B was overexpressed in both OTUD7B- sufficient and -deficient astrocytes (Fig. 6A). Overexpression of OTUD7b reduced levels of TNF-induced p-p65 (Fig. 6A) and Cxcl1, Cxcl11, Ccl20, Il6, Ccl2, and Nos2 (Fig. 6B) mRNA expression of Otud7b[fl/fl] and GFAP-cre Otud7b[fl/fl] astrocytes and completely abolished differences between the two genotypes. This confirms the critical inhibitory role of OTUD7B in TNF-induced astrocyte activation.

## OTUD7B regulates TNF signaling by sequential K63- and K48-deubiquitination of RIPK1

Since chemokine production of OTUD7B-deficient astrocytes was highest in the lesion core (Fig. 3F) and OTUD7B-regulated TNF-dependent chemokine production, we characterized the cellular source of TNF. Our spatial transcriptome data identified CD68⁺ microglia/macrophages and T cells as the major cellular source of TNF. The percentage of TNF-producing cells was similar in both Otud7b[fl/fl] and GFAP-cre Otud7b[fl/fl] mice (Fig. 7A). We next analyzed the spatial distribution of TNF mRNA in Otud7b[fl/fl] and GFAP-cre Otud7b[fl/fl] mice with EAE. In accordance with the chemokine data, expression of TNF was highest in the lesion core with a gradual decline towards the lesion rim and the peri-lesional tissue (Fig. 7B). The higher TNF levels in GFAP-cre OTUD7B[fl/fl] mice at day 15 p.i. (Fig. 7B), correspond to the increased infiltration of macrophages and T cells in the spiunal cord of GFAP-cre Otud7b[fl/fl] mice during EAE (Fig. 4B). The spatial transcriptome data also indicate that astrocytes will be exposed to TNF for several days based on the persistence of the lesions.

TNF signaling is dynamically regulated by K63 and K48 poly-ubiquitination of RIPK1. Rapidly after TNF stimulation, RIPK1 is K63 poly-ubiquitinated by TRAF2 and cIAP1 leading to RIPK1-dependent activation of NF-κB and MAPKs[29,45]. The activation of NF-κB induces expression of A20 (TNFAIP3), which cleaves K63 chains from RIPK1 and can also induce K48-dependent proteasomal degradation of RIPK1[46], which also leads to degradation of TRAF2 and termination of NF-κB and MAPK signaling[47,48]. Thus, we determined the impact of OTUD7B on RIPK1, TRAF2 and cIAP1 over a period of 5 days to reflect that under in vivo conditions astrocytes are exposed to TNF for several day. WB analysis showed that RIPK1, TRAF2 and cIAP1 protein levels remained unchanged in both genotypes but that phosphorylation of p65, p38 and JNK was increased in OTUD7B-deficient astrocytes at days 1 and 2 of TNF stimulation (Fig. 7C). From day 3 to 5, RIPK1 (Fig. 7C,) and from day 4 to 5 TRAF2 gradually declined in OTUD7B-deficient but not OTUD7B-competent astrocytes, whereas cIAP1 remained unchanged (Fig. 7C). The decline of RIPK1 protein in OTUD7B-deficient astrocytes was preceded by a strong increase of A20, which was much weaker in OTUD7B-competent astrocytes (Fig. 7C). In OTUD7B-deficient astrocytes the decline of RIPK1 and TRAF2 protein was paralleled by strong reduction of p65, p38 and JNK phosphorylation, which was not detectable in OTUD7B-expressing astrocytes (Fig. 7C).

Since RIPK1 activity and stability are regulated by its K63 and K48 poly-ubiquitination, respectively, we analyzed next whether the DUB OTUD7B regulates the RIPK1 ubiquitination status. Immunoprecipitation of RIPK1 at the various time points up to day 5 of TNF stimulation, showed that the activating K63-ubiqutination of RIPK1 was higher in

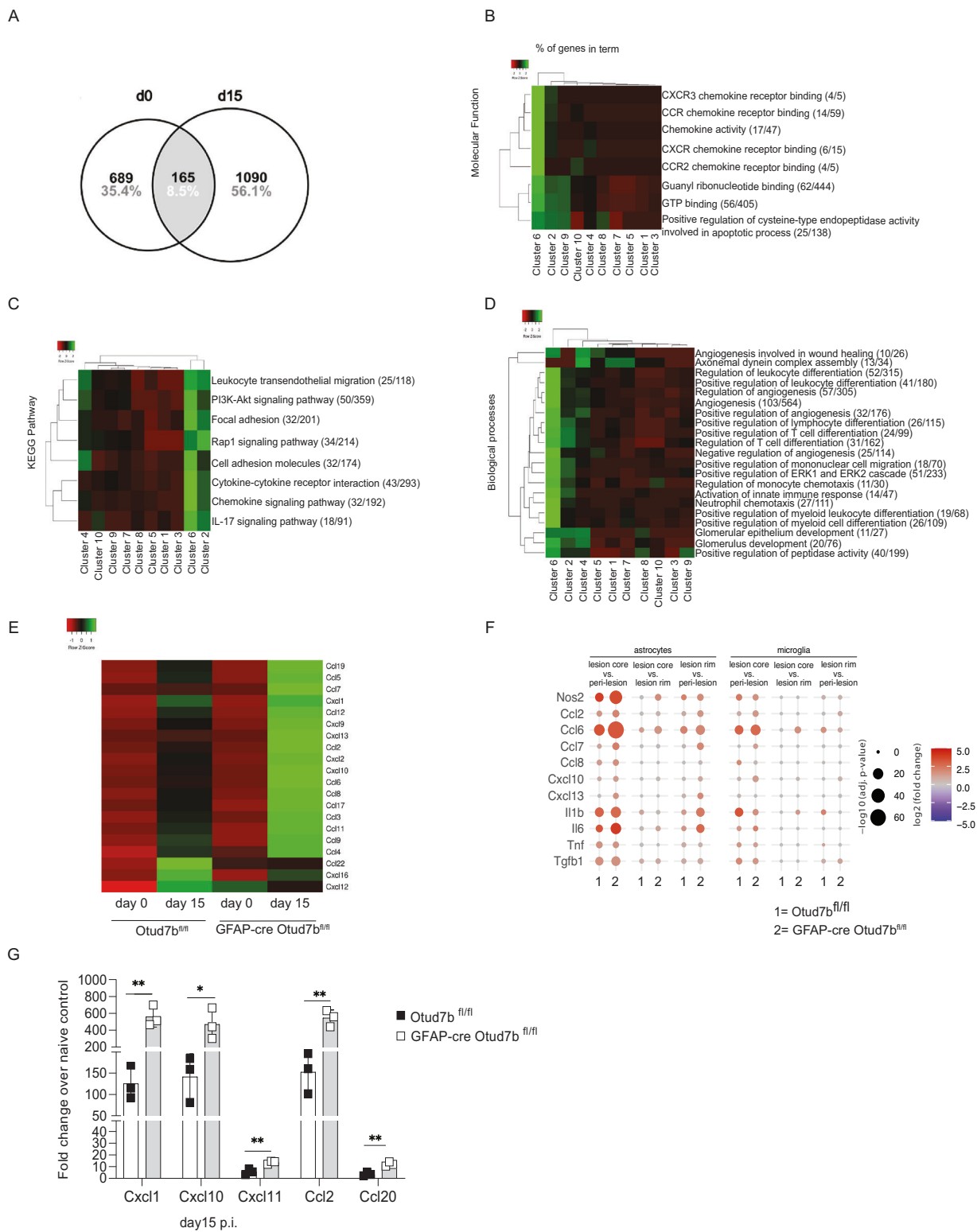

OTUD7B-deficient astrocytes at days 1 and 2 but reduced at days 3 to 5 as compared to OTUD7B-expressing astrocytes (Fig. 7D). On the contrary, K48 poly-ubiquitination of RIPK1 strongly increased from days 3 to 5 in OTUD7B-deficient but to a much lesser extent in OTUD7B-competent astrocytes (Fig. 7D). In the RIPK1 complexes, higher amounts of A20 and TRAF2 were present in OTUD7B-deficient as compared to OTUD7B-expressing astrocytes (Fig. 7D).

To validate that OTUD7B counteracts A20-mediated proteasomal degradation of RIPK1, we silenced A20 by siRNA in TNF-stimulated astrocytes. Inhibition of A20 restored RIPK1 protein in OTUD7B-deficient astrocytes to the same level as in OTUD7B-competent astrocytes (Fig. 7E). In parallel to the increased RIPK1 protein, TRAF2 increased in A20 siRNA-treated OTUD7B-deficient astrocytes (Fig. 7E), which is explained by the stabilizing role of RIPK1 for TRAF2[48]. In

**Fig. 3 | OTUD7B deficient astrocytes show pro-inflammatory gene signature.**
**A**–**F** EAE was induced in Otud7b$^{fl/fl}$ and GFAP-cre Otud7b$^{fl/fl}$ by MOG$_{35-55}$ peptide immunization combined with pertussis toxin. Spinal cord and magnetic cell sorted astrocytes were isolated from non-immunized (d0) and immunized mice at day 15 p.i. and processed for RNA sequencing. **A** Venn diagram of 1944 upregulated genes (FC < 3) in Otud7b$^{fl/fl}$ mice compared to GFAP-cre Otud7b$^{fl/fl}$ at d0 and d15 p. i. Absolute and relative values are shown. Gene ontology (**B**, **D**) and KEGG pathway (**C**) analysis was performed using ClueGO software with right-sided hypergeometric test with Bonferoni-step-down p-value correction. Only terms with FDR < 0.05 and at least GO-level 6 are shown. Numbers in brackets represent number of enriched genes compared to total number of annotated genes in that term. **E** Heat map showing the expression profile for chemokine genes from astrocytes of non-immunized and immunized mice at day 15 p. i. **F** Spatial transcriptomics analysis of differential gene expression in various lesional compartments of Otud7b$^{fl/fl}$ and GFAP-cre Otud7b$^{fl/fl}$ mice at day 15 p.i. **G** RNA was isolated from the spinal cord of non-immunized and immunized mice (day 15 p.i.). Expression of Cxcl1, Cxcl10, Cxcl11, Ccl2 and Ccl20 was analyzed by qRT-PCR. Data show fold-change increase in gene expression compared to non-immunized mice (n = 3 per group, all bars represent mean ± SEM. two-tailed Student's t-test *p < 0.05, **p < 0.01). Statistical analyses: Adjusted p-values were calculated by DEseq2 using Benjamini-Hochberg corrections of two-sided Wald test p values (**F**) and two-tailed Student's t test (**G**). Source data are provided as source data file.

addition, proteasome inhibition by MG132 prevented degradation of RIPK1 in OTUD7B-deficient astrocytes (Fig. 7F), which is in line with both the K48-dependent proteasomal degradation of RIPK1 in OTUD7B-deficient astrocytes and the prevention of proteasomal degradation by K48-deubiquitination of RIPK1 by OTUD7B in OTUD7B-competent astrocytes (Fig. 7F).

Since spatial transcriptomics revealed an increase of TNF mRNA from the healthy tissue to the core of EAE lesions (Fig. 7B), we stimulated cultivated astrocytes with increasing TNF concentrations for 5 days (Fig. 7G). These studies revealed that TNF-stimulation reduced RIPK1 proteins in OTUD7B-deficient but not in OTUD7B-competent astrocytes in dose-dependent manner (Fig. 7G). In parallel TRAF2 levels declined, whereas cIAP1 protein levels were not affected by OTUD7B-deficiency. In addition, the decrease of downstream p65, p38 and JNK phosphorylation in OTUD7B-deficient astrocytes was higher upon stimulation with increased amount of TNF (Fig. 7G). In both OTUD7B-competent and -deficient astrocytes, A20 was induced by TNF-stimulation and A20 protein levels were independent of the TNF concentration. Importantly and in agreement with the analysis of the kinetic of A20 expression (Fig. 7C), A20 levels were strongly increased in OTUD7B-deficient astrocytes, and, thus, were not affected by the reduced NF-κB activation. K48 poly-ubiquitination of RIPK1 increased in OTUD7B-deficient astrocytes stimulated with higher concentrations of TNF and this was paralleled by a TNF-dependent increase of A20/RIPK1 complex formation (Fig. 7H). Collectively, these data identify that OTUD7B regulates both K63- and K48-ubiquitination of RIPK1 in a time- and TNF dose-dependent manner.

### OTUD7B mediates GFAP-stability by its K48 deubiquitination

Increased GFAP protein expression is a hallmark of reactive astrocytes but the mechanism regulating GFAP protein levels, i.e., the impact of GFAP mRNA transcription and, in particular, GFAP protein stability and turnover are largely unresolved. Histological analysis of GFAP showed that OTUD7B was required for GFAP protein expression in EAE (Fig. 2F–H). Moreover, the reduced GFAP mRNA of GFAP-cre -Otud7b$^{fl/fl}$ mice with EAE (Fig. 2I) indicates that the reduced GFAP transcription may contribute to the reduction of GFAP protein in GFAP-cre OTUD7B$^{fl/fl}$ mice.

Since IL-6 activates STAT3 and both are important for the induction of GFAP mRNA, we analyzed first whether IL-6 mRNA production was reduced in GFAP-cre OTUD7b$^{fl/fl}$ mice with EAE. Spatial transcriptomics showed that levels of IL-6 mRNA expression in the lesions were similar between Otud7b$^{fl/fl}$ and GFAP-cre Otud7b$^{fl/fl}$ mice (Fig. 8A), indicating that differences in IL-6 production did not cause the variable GFAP mRNA expression in the two genotypes (Fig. 2I).

Next, we analyzed whether downstream of the IL-6 receptor, OTUD7B might regulate IL-6 signaling, in particular STAT3 activation. In TNF-stimulated OTUD7B-deficient astrocytes, the phosphorylation of STAT3 was increased in the first two days (Fig. 8B). and subsequently strongly reduced as compared to OTUD7B-competent astrocytes (Fig. 8B). GFAP protein and mRNA was regulated in the same kinetic with an increase in the first two days (Fig. 8B–C) and subsequent strong reduction (Fig. 8B–C). In good agreement, the reduction of pSTAT3

and GFAP was also observed in vivo in OTUD7b-deficient astrocytes (Fig. 8D), indicating that OTUD7B-dependent STAT3 phosphorylation is an important factor regulating GFAP mRNA expression. Of note, total STAT3 protein levels were OTUD7B-independent (Fig. 8B) demonstrating that OTUD7B did not regulate STAT3 stability.

To determine whether GFAP protein stability might be additionally regulated by OTUD7B, we stimulated astrocytes with TNF, inhibited IL-6, blocked new protein synthesis by CHX and prevented proteasomal degradation of proteins by MG132 treatment (Fig. 8E). We limited these experiments until day three post TNF stimulation, since from that time point onwards the impaired p38 and JNK activity of OTUD7B-deficient astrocytes might impact on GFAP mRNA expression. These experiments revealed that (i) TNF-stimulation increased GFAP protein in OTUD7B-competent but not in OTUD7B-deficient astrocytes, (ii) additional inhibition of IL-6 reduced GFAP protein in both genotypes, (iii) additional inhibition of protein synthesis by CHX further reduced GFAP and that (iv) inhibition of proteasome by MG132 restored GFAP levels in OTUD7B-deficient astrocytes to the same level as in OTUD7B-competent astrocytes (Fig. 8E). This uncovers that OTUD7B stabilizes GFAP by preventing its proteasomal degradation. In good agreement, K48 poly-ubiquitination of GFAP was strongly increased in TNF-stimulated OTUD7B-deficient astrocytes (Fig. 8F). Collectively, OTUD7B supported GFAP protein levels/abundance by two independent but synergistic mechanisms: STAT3-mediated GFAP mRNA expression based on sustained RIPK1, p38 and JNK activation leading to continued STAT3 phosphorylation and by K48-deubiquitination of GFAP critical for stabilization and increased GFAP proteins in reactive astrocytes (Fig. 9A).

## Discussion

Astrocytes are an important regulator of CNS inflammation, which can both limit and augment neuroinflammation[10,11,27,49–51]. However, the astrocyte intrinsic molecular mechanisms determining the pro- and anti-inflammatory function of astrocytes in CNS autoimmunity are incompletely understood. Analysis of public available transcriptome data of brain tissue from patients with progressive MS ([52], Fig. 1A) and our spatial transcriptome analysis of mice with EAE (Fig. 2G) identified that OTUD7B is expressed in astrocytes under homeostatic conditions and upregulated in astrocytes associated with the autoimmune inflammatory lesions. The upregulation of OTUD7B is a protective astrocyte-intrinsic mechanism ameliorating CNS autoimmunity by dynamic modulation of pro-inflammatory TNF signaling through RIPK1 deubiquitination and by upregulation of GFAP protein levels.

A hallmark of reactive astrocytes in most CNS pathologies is an increased expression of GFAP protein. Mice with OTUD7B deletion in astrocytes had normal GFAP expression under homeostatic conditions but greatly reduced or even absent GFAP protein proximal to inflammatory lesions in EAE. Mechanistically, prolonged stimulation with TNF induced direct interaction of OTUD7B with GFAP and prevented K48-dependent proteasomal degradation of GFAP. Thus, in addition to cleavage of GFAP by caspase-3[53], the deubiquitinating function of OTUD7B plays a non-redundant role for GFAP stability in reactive astrocytes.

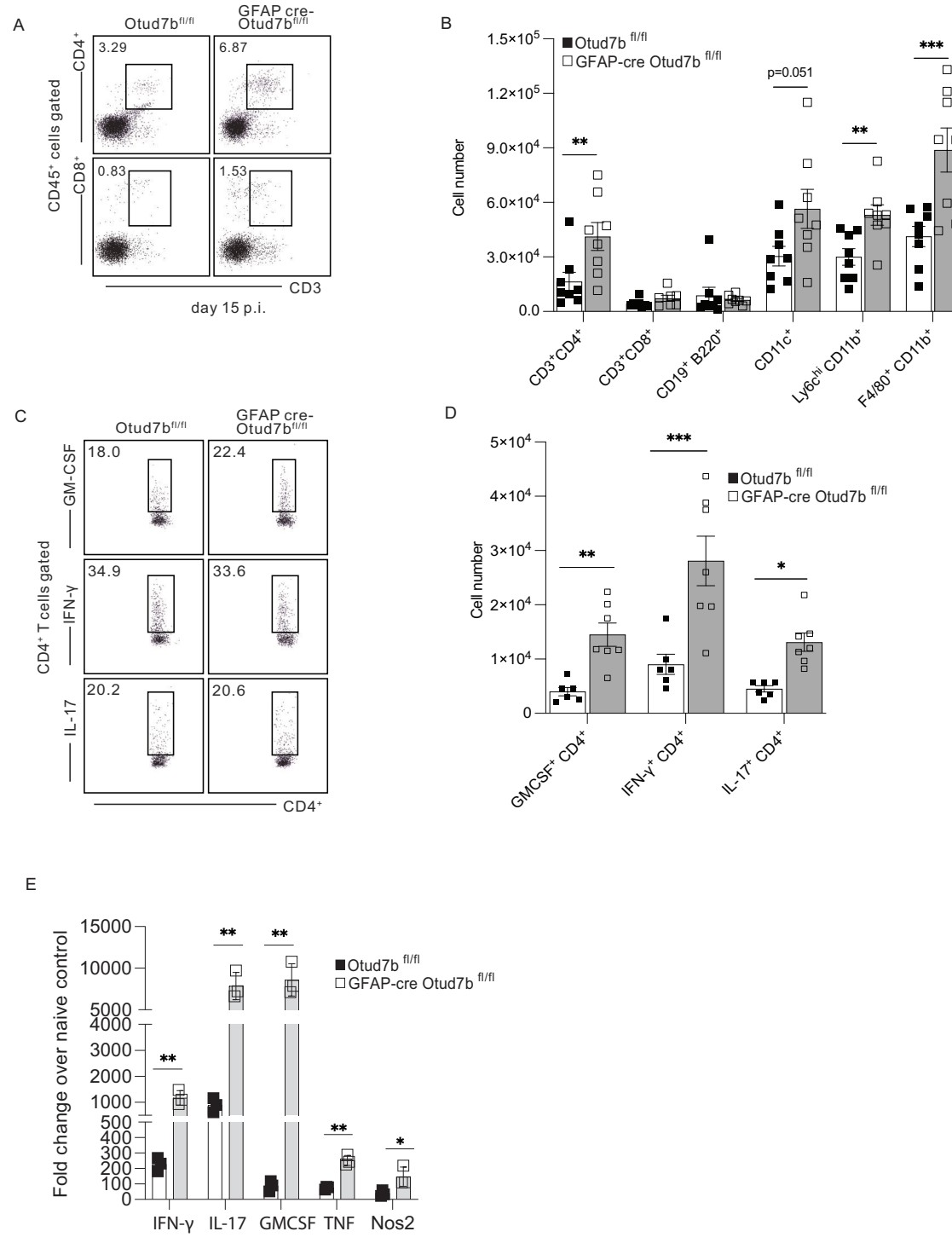

**Fig. 4 | Reduced leukocyte infiltration and inflammation in the spinal cord of Otud7b$^{fl/fl}$ mice during EAE.** EAE was induced in Otud7b$^{fl/fl}$ (n = 8) and GFAP-cre Otud7b$^{fl/fl}$ (n = 8) by MOG$_{35-55}$ peptide and spinal cord was isolated at day 0 and 15 p.i. The infiltrating leukocytes were isolated by Percoll gradient centrifugation and analyzed by flow cytometry. **A** Representative dot plots show the percentage of CD4$^+$ and CD8$^+$ T cells and (**B**) absolute number of infiltrating leukocytes in the spinal cord of Otud7b$^{fl/fl}$ and GFAP-cre Otud7b$^{fl/fl}$ mice at day15 p.i. **C, D** GM-CSF, IFN-γ and IL-17 producing CD4$^+$ T cells were analyzed by flow cytometry. Representative dot plots (**C**) and absolute numbers of GM-CSF-, IFN-γ- and IL-17-producing CD4$^+$ T cells (**D**) are shown (Otud7b$^{fl/fl}$ n = 6, GFAP-cre Otud7b$^{fl/fl}$ n = 7). **E** Relative mRNA expression of IFN-γ, IL-17, GM-CSF, TNF and Nos2 was analyzed by qRT-PCR from spinal cord of Otud7b$^{fl/fl}$ and GFAP-cre Otud7b$^{fl/fl}$ (n = 3) mice at day 15 p.i. All bars represent mean values ± SEM (**B**, **D**, **E**). Statistical analyses: two-tailed Student's t test (B,D,E). *$p < 0.05$, **$p < 0.01$, ***$p < 0.001$, ****$p < 0.0001$. ns, not significant No adjustments were made for multiple comparisons. Source data are provided as source data file.

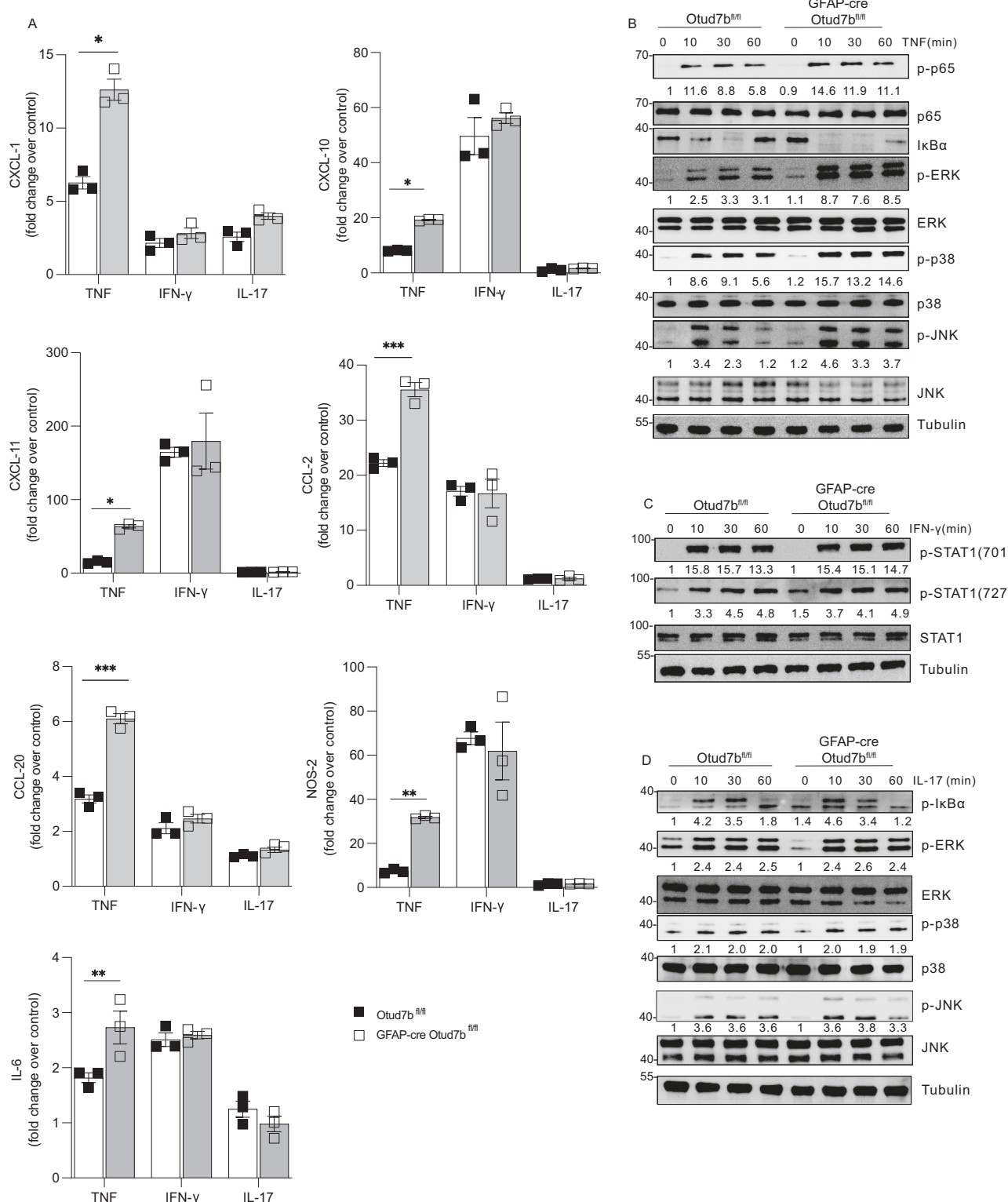

**Fig. 5 | OTUD7B impairs TNF mediated pro-inflammatory responses in astrocytes. A**, **B** Primary astrocyte cultures were prepared from P0/1 pups of Otud7b[fl/fl] and GFAP-cre Otud7b[fl/fl] mice, respectively. **A** Cells were stimulated with TNF (20 ng/mL), IFN-γ (10 ng/mL) or IL-17 (50 ng/mL) for 16 h and lysed in buffer RLT for RNA isolation. Relative mRNA expression of Cxcl-1, Cxcl-10, Cxcl-11, Ccl-1, Ccl-20, Nos-2 and IL-6 was determined by qRT-PCR (The experiment was independently repeated three times, n = 3). All graphs represent fold change over unstimulated control. All bars represent mean values ± SEM. Two-tailed student's t-test *p < 0.05, **p < 0.01, ***p < 0.001. Proteins were harvested and analyzed by WB with the indicated antibodies. Representative WB are shown after (**B**) TNF (20 ng/mL), (**C**) IFN-γ (20 ng/mL) and (**D**) IL-17 (50 ng/mL) stimulation, respectively. Blots represent one out of three independent experiments. Source data are provided as source data file.

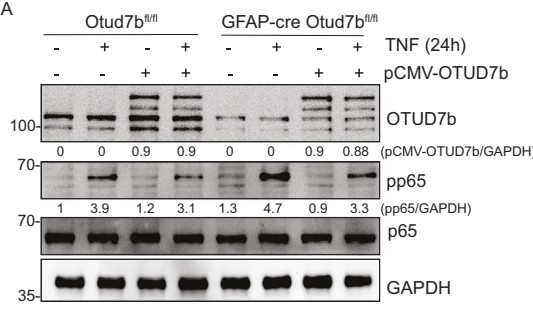

**Fig. 6 | Over-expression of OTUD7B impairs TNF mediated pro-inflammatory responses in OTUD7B deficient astrocytes. A, B** Primary astrocyte cultures were prepared from P0/1 pups of Otud7b$^{fl/fl}$ and GFAP-cre Otud7b$^{fl/fl}$ mice, respectively. OTUD7B was over-expressed in both OTUD7B-sufficient and -deficient astrocytes using the pCMV6-AC-GFP-OTUD7b expression plasmid. **A** Cells were stimulated with TNF (20 ng/mL) for 24 h and proteins were harvested and analyzed by WB with the indicated antibodies. WB representative of one of the three independent experiments. **B** Astrocytes were lysed in RLT buffer for RNA isolation. Relative mRNA expression of Cxcl-1, Cxcl-10, Cxcl-11, Ccl-1, Ccl-20, Nos-2 and IL-6 was determined by qRT-PCR (The experiment was independently repeated three times, n = 3). All graphs represent fold-change over unstimulated control. All bars represent mean values ± SEM. Statistical analyses: two-tailed Student's t test *p < 0.05, **p < 0.01, ***p < 0.001. Source data are provided as source data file.

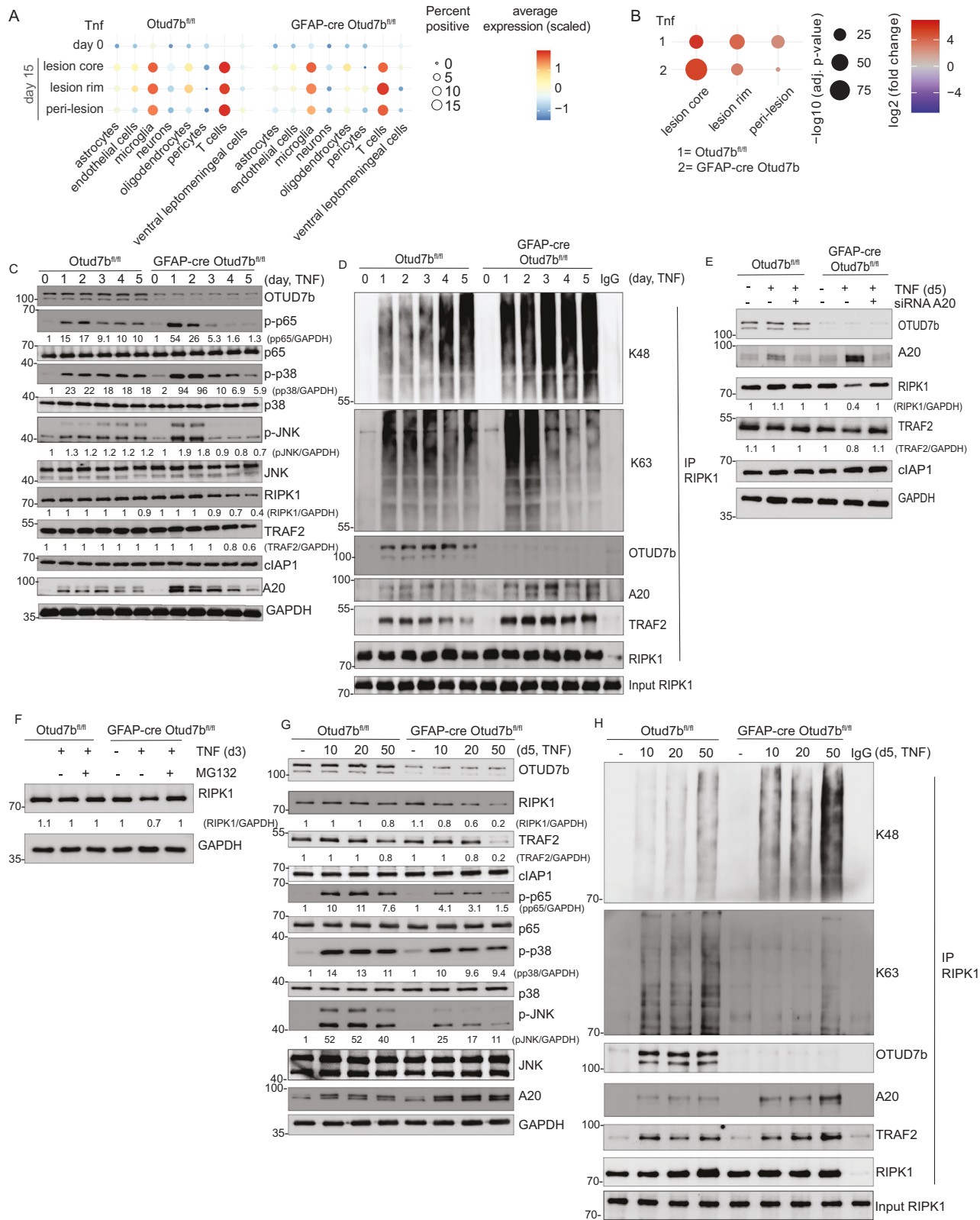

In addition, sustained GFAP mRNA expression was dependent on OTUD7B in vivo and in vitro. STAT3 is the major transcription factor inducing GFAP mRNA expression and can be activated by TNF and IL-6, respectively. Deletion of STAT3 or the cognate receptors for IL-6 family cytokines leads to strongly diminished GFAP protein expression and impaired containment of inflammatory lesions by reactive astrogliosis[5,16,54–56].

Since IL-6 mRNA levels did not differ in inflammatory lesions of OTUD7B[fl/fl] and GFAP-cre OTUD7B[fl/fl] mice but p38 and JNK were less activated due to impaired RIPK1 stability in OTUD7B-deficient astrocytes, OTUD7B might regulate GFAP mRNA expression indirectly. Key factors for the phosphorylation of STAT3 are the MAPK kinases p38 and JNK[57–62], which can be activated by RIPK1 upon TNF stimulation. OTUD7B regulated RIPK1 activity and downstream p38 and JNK

**Fig. 7 | Temporal regulation of TNF signaling by OTUD7B. A** Spatial transcriptome analysis of TNF producing cells in the spinal cord of non-immunized and immunized Otud7b[fl/fl] and GFAP-cre Otud7b[fl/fl] mice. **B** Spatial transcriptome analysis of TNF mRNA expression in EAE lesions of non-immunized and immunized Otud7b[fl/fl] and GFAP-cre Otud7b[fl/fl] mice. **C–H** OTUD7b-sufficient and -deficient primary astrocytes were stimulated with TNF (**C–F**: 20 ng/mL, **G–H** indicated concentrations). At indicated timepoints, astrocytes were lysed in RIPA lysis buffer for protein isolation. **C** Proteins were immunoblotted with the indicated antibodies. Protein expression was normalized to GAPDH and quantified based on WB data. Representative WB from one out of three experiments is shown. **D** Co-immunoprecipitation was performed with anti-RIPK1 antibody followed by immunodetection of the indicated proteins by WB. **E** OTUD7B-sufficient and -deficient astrocytes were transfected with non-specific control siRNA or A20 siRNA (5 μM) and stimulated with TNF for 5 days. At day 5 post-stimulation, astrocytes were lysed

with RIPA buffer and proteins were analyzed by WB using indicated antibodies. RIPK1 and TRAF2 expression was normalized to GAPDH and quantified based on WB data. **F** Astrocytes were treated with TNF for 3 days and MG132 was added on day 3 for 6 hours to inhibit the proteasome. Astrocytes were lysed in RIPA buffer and the protein lysates were analyzed by WB for RIPK1 and GAPDH. **G, H** OTUD7B-sufficient and -deficient primary astrocytes were stimulated with 10 ng/mL, 20 ng/mL, and 50 ng/mL TNF for 5 days. On day 5 post-stimulation, cells were lysed in RIPA lysis buffer and proteins were isolated for (**G**) analysis of protein expression by WB and (**H**) identification of interacting proteins by co-immunoprecipitation of protein complexes with anti-RIPK1 antibody and WB. All WB blots are representative of one of three independent experiments. Statistical analyses: Adjusted p-values were calculated by DEseq2 using Benjamini-Hochberg corrections of two-sided Wald test p values (**B**). Source data are provided as source data file.

activation (Fig. 6B). Thus, OTUD7B might regulate GFAP mRNA expression indirectly via the RIPK1-p38/JNK-STAT3 axis.

In EAE and other inflammatory diseases of the CNS, reactive astrocytes can form borders around inflammatory lesions, which contributes to the local restriction of the neuroinflammation and CNS damage[63–65]. Studies in GFAP-deficient mice revealed that deletion of GFAP leads to a more widespread inflammation and more severe disease in EAE and bacterial and parasitic CNS infections[63,66–68]. This contributed to the concept that bordering of inflammatory lesions by reactive astrocytes with increased GFAP expression contributes to the local containment of inflammation (reviewed by Sofroniew[9]). Of note, a low level of K48 ubiquitination and GFAP degradation were also detected in OTUD7B-competent astrocytes (Fig. 8E). This may be an important mechanism to prevent pathological GFAP accumulation in reactive astrocytes. Excessive accumulation of GFAP in astrocytes induced by gain-of-function mutations in the GFAP gene and by impaired proteasomal GFAP degradation is toxic and underlies human Alexander disease and its corresponding mouse models[69]. Perspectival, it would be interesting to explore OTUD7B in this astrocytopathy and to determine whether GFAP degradation induced by OTUD7B inhibition might have a therapeutic effect. In addition, it remains to be determined which E3 ligases mediate K48 ubiquitination of GFAP. Collectively, these data imply that the regulation of GFAP abundance by OTUD7B-dependent deubiquitination is an important general mechanism potentially regulating the function of reactive astrocytes independent of the underlying disease.

In addition to reduced GFAP protein, the dominant phenotype of OTUD7B-deficient astrocytes was increased chemokine production in the core and rim of inflammatory EAE lesions, which was associated with an increased recruitment of encephalitogenic T cells, more widespread inflammation and demyelination. Detailed analysis of the molecular functions of OTUD7B identified that OTUD7B rapidly interacted with RIPK1 upon TNF exposure. RIPK1 is a central signaling molecule regulating the activation of pro-inflammatory NF-κB and MAPK signaling as well as cell death pathways[70,71]. The ubiquitination status of RIPK1 is critical to induce or inhibit these individual cellular pathways[29,45]. OTUD7B limited NF-κB and MAPK activation by reducing RIPK1 K63-ubiquitination, which is important for NF-κB - and MAPK-dependent cytokine and chemokine production of TNF-stimulated astrocytes. The reduced chemo- and cytokine expression of OTUD7B-competent astrocytes was evident in vitro, in the bulk RNAseq analysis of ex vivo isolated astrocytes and spatial transcriptomic analysis of mice with EAE. In TNF activated cells, NF-κB activation leads to the expression of A20 which subsequently interacts with and inhibits RIPK1. In this negative feedback loop, A20 inhibits sustained RIPK1 activation by RIPK1 K63-deubiquitination and by inducing K48-dependent proteasomal degradation of RIPK1. Here, we identified that the A20-mediated dynamic change of RIPK1 ubiquitination, resulted in a shift of the targeted ubiquitin chains of OTUD7B from K63 to K48 of RIPK1, and that OTUD7B interacted with

A20 and diminished K48 polyubiquitination of RIPK1. In the absence of OTUD7B, A20 induced K48 ubiquitination of RIPK1 induced its proteasomal degradation. Functionally important and in agreement with the present study deletion of A20 in astrocytes results in an augmented and sustained NF-κB and also STAT1 activation leading to aggravation of EAE[27].

Of note, the dominant in vivo phenotype of OTUD7B-deficient astrocytes was increased activation and chemokine production at day 15 p.i., i.e., when clinical symptoms of EAE already existed for several days. Thus, the OTUD7B-regulated in vitro shift from TNF-induced increased to reduced astrocyte activation was not detectable in vivo. In this regard, it should be stressed that (i) other NF-κB and MAPK activating pathways, in particular IL-17 signaling, were not regulated by OTUD7B (Fig. 5D), and (ii) IL-17 was increased expressed in the CNS of GFAP-cre Otud7b[fl/fl] mice (Fig. 4E). In addition, disturbance of GFAP protein expression induces endoplasmic reticulum stress, increased activation of MAPK kinases and neuroinflammation as observed in murine models of Alexanders diseases[69]. Thus, the in vitro diminished RIPK1 signaling upon prolonged TNF exposure might be in vivo compensated by other pro-inflammatory signaling pathways contributing to the sustained increased chemokine production by astrocytes.

TNF can induce RIPK1-dependent cell death, if K63 ubiquitination of RIPK1 is impaired. Histologically, the greatly diminished GFAP protein expression of OTUD7B-deficient astrocytes gave the impression that these cells might have been eliminated by apoptosis. However, we detected identical presence of SOX2[+] Sox9[+] astrocytes in both GFAP-cre Otud7b[fl/fl] and Otud7b[fl/fl] mice and spatial transcriptomics showed that astrocyte numbers did not differ in the core, rim and peri-lesion between the two genotypes. Thus, OTUD7B-deficient astrocytes with impaired K63-ubiquitination of RIPK1 were highly resilient against TNF-induced cell death. This is in contrast to OTUD7B-deficient dendritic cells, which rapidly undergo apoptosis upon exposure to TNF in murine cerebral malaria[29]. On the contrary, OTUD7B does not regulate apoptosis in T cells but facilitates proximal T cell receptor signaling by deubiquitination of the tyrosine kinase ZAP70 in EAE and murine listeriosis, two diseases characterized by the production of large amounts of TNF. At present, the cell type-specific differences underlying the differential impact of OTUD7B on TNF induced apoptosis are unresolved. Our current study on OTUD7B expands our previous understanding on the regulation of astrocyte reactivity by the DUBs A20[27] and OTUB1[10]. All three DUBs are negative regulators of astrocyte-mediated inflammation in EAE, though via distinct mechanisms. A20 restricts NF-κB- and STAT1-mediated chemokine production by astrocytes in response to TNF, IL-17, and IFN-γ, respectively. Meanwhile, OTUB1 suppresses IFN-γ-driven astrocyte activation by deubiquitinating the suppressor of cytokine signaling SOCS1. In contrast, OTUD7B regulates both inflammatory and structural astrocyte functions by sequential K63- and K48-deubiquitination of RIPK1 and by directly stabilizing GFAP protein through K48-deubiquitination, respectively. Together, these findings highlight a critical and non-

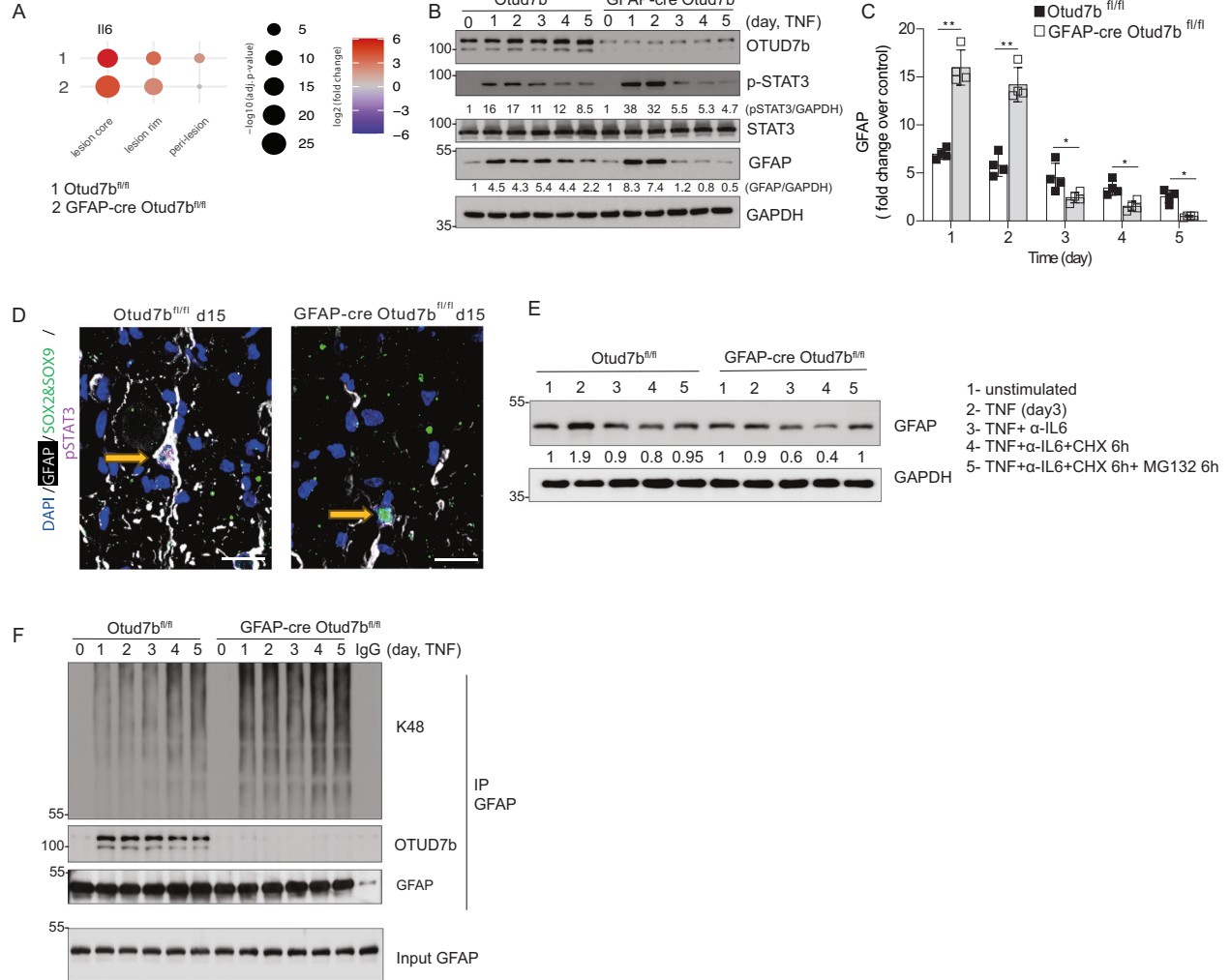

**Fig. 8 | OTUD7B prevents K48-ubiquitination of GFAP and its subsequent proteasomal degradation. A** Spatial transcriptome analysis of IL-6 mRNA expression in EAE lesions of non-immunized and immunized Otud7b^fl/fl and GFAP-cre Otud7b^fl/fl mice. **B, C** Primary astrocytes were stimulated with 50 ng/mL of TNF and lysed at the indicated time points in RIPA lysis buffer for protein isolation (**B**, **F**) or RLT buffer for RNA extraction (**C**). In (**B**), WB was performed to analyze the expression levels of OTUD7B, p-STAT3, STAT3, GFAP, and GAPDH. Protein expression was normalized to GAPDH and quantified based on WB data. Representative WB from one out of three experiments is shown. **C** Fold-change in mRNA expression levels of *gfap* over unstimulated control was analyzed by qRT-PCR and normalized to *hprt*. (The experiment was independently repeated three times, n = 3) **D** Immunofluorescence staining showing SOX2/SOX9⁺GFAP⁺ pSTAT3⁺ astrocytes in the spinal cord of Otud7b^fl/fl, while SOX2/SOX9⁺ astrocytes of GFAP-cre Otud7b^fl/fl mice are largely GFAP- and pSTAT3-negative at day 15 p.i. (Scale bars =

10 μm). **E** OTUD7B-sufficient and -deficient astrocytes were stimulated with 20 ng/mL of TNF alone, or in combination with anti-IL6 neutralizing antibody for 3 days. On day 3 post-stimulation, 10 μg/mL of CHX and/or 10 μM of MG132 were added to the cells for 6 hours, and proteins were harvested for WB. The expression level of GFAP was analyzed by WB using GAPDH as a loading control. Relative protein expression of GFAP normalized to GAPDH is shown. **F** Protein complexes from total protein lysates were immunoprecipitated using anti-GFAP antibody, and precipitates were analyzed for GFAP, OTUD7B, and K48 ubiquitin chains by WB. All graphs represent fold-change over unstimulated control. All bars represent mean values ± SEM. Statistical analyses: Statistical analyses: Adjusted p-values were calculated by DEseq2 using Benjamini–Hochberg corrections of two-sided Wald test p values (**A**) two-tailed Student's t test (**C**) *p < 0.05, **p < 0.01, ***p < 0.001. Source data are provided as source data file.

redundant role of DUBs in controlling astrocyte reactivity and CNS autoimmunity.

This study illustrates that astrocyte activity is dynamically and interdependently regulated by the concentration of external factors such as TNF and by intrinsic signaling molecules, in particular OTUD7B. Although OTUD7B is a critical factor regulating astrocyte reactivity in EAE and unique in its capacity to regulate both GFAP and RIPK1, the interplay with other external factors including those derived from the microbiome, which also regulate astrocyte intrinsic responses[72–75], still has to be explored. Thus, it remains to be determined whether OTUD7B has the same function and central role under other experimental conditions and in human CNS disorders. At present the complex network of interdepend external and intrinsic factors

regulating astrocyte reactivity cannot be resolved in its kinetic at the spatial level for single astrocytes. Thus, it remains to be determined whether OTUD7B regulates the plasticity of astrocytes or is a determinant for the development of specific astrocyte subpopulations under inflammatory conditions.

## Methods
### Mice
OTUD7B^fl/fl mice with C57BL/6 background were generated with C57BL/6N-^Otud7btm1b(EUCOMM)Wtsi/Wtsi embryonic stem cells purchased from the European Mouse Mutant Archive. First, *Otud7b* mutant mice were crossed with B6.129S4-*Gt(ROSA)26Sor*^tm1(FLP1)Dym/RainJ (Stock No: 009086, The Jackson Laboratory, Bar Habor, ME, USA) to delete the *frt-*

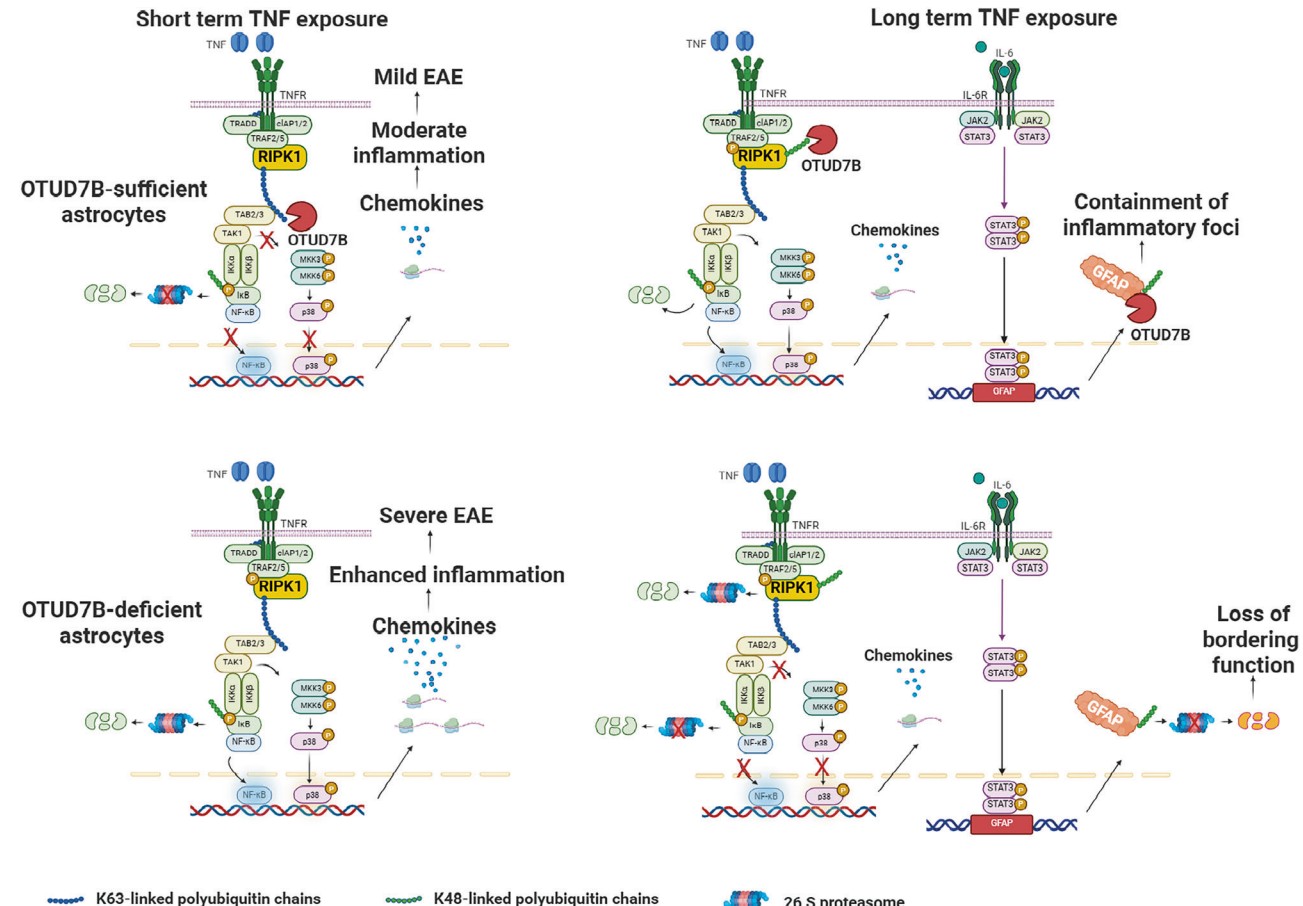

**Fig. 9 | Schematic representation showing the dual protective function of astrocytic OTUD7B in EAE.** OTUD7B (i) prevents enhanced inflammation by limiting TNF-induced chemokine production in astrocytes through early K63- and late K48-deubiquitination of RIPK1, and (ii) fosters astrocyte bordering of lesions by preventing proteasomal degradation of GFAP through K48-deubiquitination. The figure is created in BioRender. Gopala Nishanth. (2025) https://BioRender.com/rin8c1c.

flanked sequences. Thereafter, *otud7b* mutants were crossed with C57BL/6 GFAP-cre mice[33] to obtain GFAP-cre Otud7b$^{fl/fl}$ mice. The genotyping was performed by PCR of the tail DNA with primers specific for GFAP (GFAP Cre Sense 5′–GACACCAGACCAACTGG TAATGGTAGCGAC–3, GFAP Anti-Sense 5′–GCATCGAGCTGGGTAAT AAGCGTTGGCAAT–3′) and Otud7b flox (Otud7b Sense 5′–CAG AATAAAGAGGTGGGTGAGC–3′, Otud7b Anti-Sense 5′–CACATTGCAG TGATCATGC–3′) respectively. Wildtype C57BL/6 mice were obtained from Janvier (Le Genest-Saint Isle, France). Animals were kept under specific pathogen-free (SPF) conditions in animal facilities of the Otto-von-Guericke University Magdeburg (Magdeburg, Germany) and Hannover Medical School (Hannover, Germany). Animal care and experimental procedures were carried out according to the European animal protection law and approved by local authorities (Landesverwaltungsamt Halle, file number 42502-2-1260).

### Induction of EAE and clinical assessment
For active EAE induction, 8–12 weeks old mice were immunized with 200 µg of myelin oligodendrocyte glycoprotein (MOG)$_{35-55}$ peptide mixed in complete Freund's adjuvant containing 800 µg of killed *Mycobacterium tuberculosis*. In addition, 200 ng pertussis toxin dissolved in 200 µl PBS was intraperitoneally injected respectively at day 0 and 2 post immunization (p.i.). The symptoms and body weight were monitored daily in a double-blinded way according to a previously published score with increasing severity from 0 to 5 as follows[10]: 0, no signs; 0.5, partial tail weakness; 1, limp tail or slight slowing of righting from supine position; 1.5, limp tail and slowing of righting; 2, partial

hind limb weakness; 2.5, dragging of hind limb(s) without complete paralysis; 3, complete paralysis of at least one hind limb; 3.5, hind limb paralysis and slight weakness of forelimbs; 4, severe forelimb weakness; 5, moribund or dead. Daily clinical scores were displayed as the mean of all individual disease scores within each group. Mice were euthanized using ketamine and xylazine. Both male and female mice were used in this study

### Astrocyte isolation from adult mice
Spinal cords were isolated from anaesthetized (ketamine and xylazine) and PBS-perfused naïve and EAE mice at day 15 d p.i. Single-cell suspension was generated using NeuroCult™ Enzymatic Dissociation Kit according to the manufacturer's instruction. Astrocytes were purified from the single-cell suspension with the anti-ACSA-2 Microbead Kit and the purity was analyzed by flow cytometry with anti-ACSA-2-PE antibody.

### Isolation of cells from the spinal cord and flow cytometry
To obtain leukocytes from the spinal cord of non-immunized mice (d0) and mice with EAE (d15 p.i.) mice were first cardially perfused with 0.1 M PBS (pH 7.4) in deep ketamine and xylazine anesthesia. Immediately thereafter, spinal cords were removed, minced through 70µm cell strainers followed by Percoll® gradient centrifugation. Cells were counted with a hemocytometer and stained with fluorochrome-coupled antibodies against CD4, CD3, CD45, CD8, CD19, B220, Ly6C, Ly6G, CD11b and CD11c to differentiate T cells, B cells, inflammatory monocytes, granulocytes, macrophages, and dendritic cells

respectively (see Supplementary dataset 2). For intracellular staining, cells were incubated with PMA (50 ng/ml), ionomycin (500 ng/ml), and Brefeldin A (1 µg/ml) in RPMI 1640 medium supplemented with 10% FCS, 1% Non-essential amino acids (NEAA), and 1% L-glutamine at 37 °C for 4 h. Thereafter, cells were stained with CD3, CD4, CD45 antibodies, fixed and permeabilized with Intracellular Fixation/Permeabilization Kit followed by staining with anti-IL-17, anti-GM-CSF, and anti-IFN-γ antibodies, respectively. Flow cytometry was performed on a Cytek Northern Light Flow Cytometer and data were analyzed with the FlowJo software.

### Histology

Mice anesthetized using ketamine and xylazine were perfused with 0.1 M PBS followed by 4 % paraformaldehyde in PBS. After embedding in paraffin, sections of brains and spinal cords were used for hematoxylin & eosin and cresyl violet-luxol fast blue (CV-LFB) staining. Expression of GFAP was demonstrated in an ABC protocol with 3,3' diaminobenzidine and $H_2O_2$ as substrate. For the immunofluorescent staining, the slides were deparaffinized for 1 h at 60 °C, followed by a removal of the residual paraffin with 2 × 15 min washing in xylene. Next, the tissue was rehydrated and incubated in 0.5% NaBH4 for 30 min at room temperature to reduce autofluorescence[76]. For antigen retrieval, the slides were incubated in 10 mM citric acid buffer (with 2 mM EDTA, 0.05% Tween20) for 15 min at 95 °C and allowed to cool down for 20 min. After washing in PBS-T (1% TritonX-100) for 30 min and PBS for 10 min at room temperature, slices were incubated with the primary antibodies diluted in blocking solution (3% NDS, 0.5% TritonX-100) at 4 °C[77]. As primary antibodies rabbit anti-GFAP (1:1000), mouse anti-Sox2 (1:500) goat anti-SOX9 (1:500) rat anti-Iba-I (1:800) rabbit anti-OTUD7b (1:1000) rabbit anti-PSTAT3 (1:1000) and rabbit anti-GFAP (1:1000) were used. After 2 × 15 min washing with PBS the slices were incubated with secondary antibodies diluted in blocking solution at 4 °C overnight. As secondary antibodies Alexa488-conjugated donkey anti-goat (1:400), Alexa488-conjugated donkey anti-mouse (1:400), Cy3-conjugated donkey anti-rat (1:400) and Cy5-conjugated donkey anti-rabbit (1:400) were used. Subsequently, secondary antibodies were removed, and nuclei were stained using DAPI (1:10000). After 3 × 10 min washing in PBS, slides were mounted with Aquapolymount solution and stored at 4 °C. Images were taken using a Zeiss inverted Axio Observer seven with ApoTome.2 equipped with an Axiocam 503 and a Colibri 7 LED light source, and Zeiss LSM 780 with four lasers (405, 488, 559 and 633 nm) and ×20, ×40 and ×63 objective lenses. For apotome acquisition on Zen2.6 pro software was used.

### Primary astrocyte cultures and treatment

Primary astrocytes were isolated from 1- to 2-day-old newborn mice and cultured in DMEM containing 1% glutamine, 10% FCS, and 1% penicillin/streptomycin as described before[10]. The purity of astrocyte cultures was more than 95%, as assessed by flow cytometry with antibodies against CD11b and ACSA-2. For the analysis of cytokine receptor-activated signaling pathways, astrocytes were stimulated at the indicated concentrations with TNF (10 ng/ml), IL-17 (50 ng/ml), and IFN-γ (10 ng/ml), respectively, for the indicated time points. For long term TNF treatment, primary astrocytes were stimulated with increasing concentrations of TNF (10 ng/mL, 20 ng/mL or 50 ng/mL) for 5 days.

### CHX chase assay

To detect the stability of GFAP protein, cells were stimulated with either TNF alone (50 ng/mL) or in combination with α-IL6 antibody (2 µg/mL) for 3 days. At d3 post-stimulation, cells were treated with 10 µg/mL of cycloheximide (CHX) with/without 10 µM of proteasome inhibitor MG132 for 6 h.

### siRNA transfection

For siRNA-mediated knockdown of A20, primary astrocytes were transfected with 5 µM of A20-specific siRNA according to the manufacturer's instructions. Thereafter, the cells were stimulated with 50 ng/mL of TNF for 5 days, followed by protein isolation and WB analysis

### Western blot

Samples from primary astrocytes and mouse organs were lysed on ice in RIPA lysis buffer supplemented with PhosSTOP, phenylmethylsulfonyl fluoride (PSMF) and protease inhibitor cocktail. Cell lysates were pre-cleared by centrifugation at 10,000 g for 15 min at 4 °C. Supernatant was collected and quantified by BCA assay according to manufacturer's protocol. Protein samples heated in lane marker reducing sample buffer at 99 °C for 5 min. Equal amounts of samples were separated by SDS-PAGE and subsequently transferred to polyvinylidene difluoride (PVDF) membranes, which were blocked with 5% BSA at room temperature for 1 h, followed by incubation with mentioned primary antibodies (Supplementary dataset 2) at 4 °C overnight.

Blots were developed using the ECL Plus Kit and images were captured on Intas Chemo Cam Luminescent Image Analysis system (INTAS). Quantification and analysis of WB images was performed with the LabImage 1D software.

### Co-immunoprecipitation (Co-IP)

Whole cell lysates from astrocytes were precleared by incubation with GammaBind G Sepharose beads with gentle shaking at 4 °C for 2 h. After removal of beads by centrifugation, samples were incubated with specific antibodies under continuous shaking at 4 °C overnight. Following day, antibody-protein complexes were captured by incubating samples with GammaBind G Sepharose beads at 4 °C for 2 hours. Thereafter, the beads were washed with ice cold PBS thrice, resuspended in 2x lane marker reducing sample buffer and boiled at 99 °C for 5 min. Samples were centrifuged and supernatant was collected for WB analysis.

### Quantitative RT-PCR

Total mRNA was isolated from spinal cord tissue or astrocytes in buffer RLT using the RNeasy Mini Kit according to manufacturer's protocol. mRNA was reverse-transcribed into cDNA with the SuperScript Reverse Transcriptase Kit. Quantitative RT-PCR was performed with a LightCycler 480 system using TaqMan probes (Supplementary dataset 2). Gene expression levels were normalized to internal control *Hprt* and fold change increase in gene expression over naïve controls was calculated according to the ΔΔ cycle threshold (CT) method (Livak and Schmittgen, 2001[78]).

### Transcriptome analysis of isolated astrocytes

Astrocytes were isolated by magnetic microbeads from spinal cords of Otud7b^fl/fl and GFAP-cre Otud7b^fl/fl mice, respectively, at day 15 p.i. Astrocytes isolated from naïve Otud7b^fl/fl and GFAP-cre Otud7b^fl/fl mice were used as control. For each experimental group, astrocytes from three mice (n = 3) were pooled to reduce biological variability, resulting in one sample per condition. RNA was isolated using RNeasy Mini Kit (Qiagen). RNA quality and integrity of total RNA was controlled on Agilent Technologies 2100 Bioanalyzer (Agilent Technologies). The RNA sequencing library was generated from 500 ng total RNA using Dynabeads mRNA DIRECT Micro Purification Kit (Thermo Fisher) for mRNA purification followed by ScriptSeq v2 RNA-Seq Library Preparation Kit (Epicenter) according to manufacture´s protocols. The libraries were sequenced on Illumina HiSeq2500 using TruSeq SBS Kit v3-HS (50 cycles, single ended run) with an average of $3 \times 10^7$ reads per RNA sample. FASTQ data quality was evaluated by FASTQC tool. Before

alignment to reference genome each sequence in the raw FASTQ files were trimmed on base call quality and sequencing adapter contamination using Trim Galore. Reads shorter than 20 bp were removed from FASTQ file. Trimmed reads were aligned to the reference genome using open source short read aligner STAR. Feature counts were determined using R package Rsubread. Only genes showing counts greater than 5 at least two times across all samples were considered for further analysis. Gene annotation was done by R package bioMaRt. Data normalization was performed by calculating log2 counts per million (CPM) values and quantile normalization (edgeR). Based on the 20th percentile of distribution of the normalized log2 CPM values a CPM cutoff of 0.56 was derived to discriminate lowly abundant transcripts. Only transcripts with log2 CPM > 0.56 in at least one sample were used for further analysis. Differential expression between samples from Otud7b knockout versus Otud7b-competent mice isolated on day 0 and day 15 was calculated, respectively. Transcripts with fold change (FC) with |FC| > 3 were regarded relevant.

The online platform DAVID Bioinformatics from National Institutes of Health (NIH) was used for the Gene Ontology and KEGG pathway enrichment analysis.

### Spatial transcriptomics

In situ RNA expression analysis at a single-cell level was performed on spinal cords of Otud7b$^{fl/fl}$ and GFAP-cre Otud7b$^{fl/fl}$ mice, respectively, at day 15 p.i. using the Xenium system (10x Genomics). Astrocytes isolated from naïve Otud7b$^{fl/fl}$ and GFAP-cre Otud7b$^{fl/fl}$ mice were used as control (n = 3). 5 μm thick sections were placed on a Xenium slide according to the manufacturer's protocol, with drying at 42 °C for 3 hours and overnight placement in a desiccator at room temperature, followed by deparaffinization and permeabilization to make the mRNA accessible. The Probe Hybridization Mix was prepared using a pre-designed panel with 247 genes (Xenium Mouse Brain Gene Expression Panel v1) and custom add-on panel with 50 genes (Xenium Custom Gene Expression panel, design ID: QZD68C) according to the user guide (CG000582, Rev D, 10x Genomics). The staining for Xenium was performed using Xenium Nuclei Staining Buffer (10x Genomics product number: 2000762) as a part of the Xenium Slides & Sample Prep Reagents Kit (PN-1000460). Following the Xenium run, Hematoxylin and Eosin (H&E) staining was performed on the same section according to the Post-Xenium Analyzer H&E Staining user guide (CG000613, Rev B, 10x Genomics). Spatial transcriptome data samples were processed using the VoltRon package (GitHub - LandthalerLab/EAE_Otud7b). Differential expression was calculated using DESeq2 based on "pseudobulk" raw counts (i.e., per cell type and per sample, counts are added up for all cells). Count normalization per gene was done using the formula log((raw count for the gene)/(raw count sum for the cell) * (size factor) +1), with a size factor of 10000, and using natural logarithms.

### Transfection of primary astrocytes

Primary astrocytes were transfected with pCMV6-AC-GFP-OTUD7b expression plasmid using Lipofectamine 3000 (Thermo Fisher) according to the manufacturer's instruction. After 48 hrs post transfection, cells were stimulated with 20 ng of TNF for 24 hr, followed by protein isolation and western blot analysis as described before.

### Quantification and statistical analysis

Quantification and analysis of WB images was performed with the LabImage 1D software. Statistical significance was determined by two-tailed $t$-tests using Prism v.10 (GraphPad) with p > 0.05=ns (not significant), *p ≤ 0.05, **p ≤ 0.01 and ***p ≤ 0.001. Error bars in the figure represent standard error of mean (SEM). N indicates the number of biological replicates. For spatial transcriptomics analysis adjusted p-values were calculated by DESeq2 using Benjamini-Hochberg corrections of two-sided Wald test p values.

### Reporting summary

Further information on research design is available in the Nature Portfolio Reporting Summary linked to this article.

## Data availability

Spatial transcriptomics raw data generated in this study have been deposited in Gene Expression Omnibus (GEO) under the accession number GSE286422. Bulk RNA-sequencing data have been submitted under accession number GSE286263. Data The public RNA-seq used in this study is available in the GSE180759. Source data are provided with the paper. Source data are provided with this paper.

## Code availability

The code generated for this study is available at https://doi.org/10.5281/zenodo.16950102.

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

## Acknowledgements

The authors thank Birgit Brennecke, Kerstin Ellrott, Elena Fischer, Izabela Plumbon and Nadja Schlüter for expert technical assistance. This work was supported by a grant from the DFG (CRC 854, project A5 to DS) and by DFG EXC 2155 "RESIST" (Project ID39087428 to DS).

## Author contributions

D.S., G.N., and X.W. conceptualized the study and supervised the experiments. K.H., W.Y., A.J., J.S., RB., E.W., A.M., M.D., H.R., T.C., J.A., and M.L. performed the experiments and analyzed the data. K.H, D.S., G.N., and X.W. interpreted the data. D.S. and G.N. wrote the manuscript.

## Funding

## Competing interests

The authors declare no competing interests.

## Additional information

[1]Institute of Medical Microbiology and Hospital Epidemiology, Hannover Medical School, Hannover, Germany. [2]Institute of Medical Microbiology and Hospital Hygiene, Otto-von-Guericke University Magdeburg, Magdeburg, Germany. [3]Institute of Biochemistry, Friedrich-Alexander-Universität Erlangen-Nürnberg, Erlangen, Germany. [4]Berlin Institute for Medical Systems Biology (BIMSB), Max-Delbrück-Center for Molecular Medicine in the Helmholtz Association (MDC), Berlin, Germany. [5]Department of Neuropathology, Faculty of Medicine and University Hospital Cologne, University of Cologne, Cologne, Germany. [6]Institute of Neuropathology, Charité—University Medicine Berlin, Institute of Neuropathology, Berlin, Germany. [7]Genomics Technology Platform, Max Delbrück Center for Molecular Medicine, Berlin, Germany. [8]Berlin Institute of Health at Charité, Berlin, Germany. [9]Institut für Biologie, Humboldt-Universität zu Berlin, Berlin, Germany. [10]Oujiang Laboratory (Zhejiang Lab for Regenerative Medicine, Vision and Brain Health), School of Pharmaceutical Sciences, Wenzhou Medical University, Wenzhou, China. [11]Cluster of Excellence-Resolving Infection Susceptibility (RESIST), (EXC 2155), Hannover Medical School, Hannover, Germany. [12]These authors contributed equally: Xu Wang, Gopala Nishanth, Dirk Schlüter. ✉e-mail: Schlueter.dirk@mh-hannover.de

