## [Transparent Peer review file · Nature Communications]

Astrocytic-OTUD7B ameliorates murine experimental autoimmune encephalomyelitis by stabilizing glial fibrillary acidic protein and preventing inflammation

Corresponding Author: Professor Dirk Schlüter

Version 0:

Reviewer comments:

Reviewer #1

(Remarks to the Author)

1. Key Results

Reviewer's Overview:

In their manuscript "The deubiquitinase 1 OTUD7B ameliorates central nervous system autoimmunity by inhibiting degradation of glial fibrillary acidic protein and astrocyte hyperinflammation", Harit and colleagues demonstrate that OTUD7B is upregulated in human and murine inflammatory CNS disorders. They show that deletion of OTUD7B in astrocytes (using GFAP-cre *Otud7b^{fl/fl}* mice) leads to more severe EAE with altered astrocyte morphology, reduced GFAP expression, and enhanced cytokine/chemokine production that in turn increases lymphocytic and monocytic recruitment into the CNS.

2. Validity

Lines 183–222 (Astrocyte-specific deletion and in vivo validation):

The generation and validation of the GFAP-cre *Otud7b^{fl/fl}* mouse model are clearly presented. The in vivo data support the claim that loss of OTUD7B in astrocytes exacerbates EAE pathology.

Lines 225–285 (Chemokine production and leukocyte infiltration):

Data showing increased chemokine mRNA levels and enhanced recruitment of encephalitogenic CD4⁺ T cells further support the authors' conclusions regarding the regulatory role of OTUD7B.

Lines 287–374 (TNF signaling analyses):

Detailed biochemical assays illustrate that OTUD7B regulates TNF-induced signaling by modulating the ubiquitination state of RIPK1. This mechanistic insight is critical for understanding the downstream effects on NF-κB activation.

Reviewer's Comment:

Although the experimental design is robust overall, additional clarity on replicate numbers and data normalization—especially for the RNA-seq and spatial transcriptomics experiments—is needed to further reinforce the data's robustness. Moreover, the reviewer questions whether OTUD7B is truly acting as a regulator of neuroinflammation (as claimed in lines 180/181) or simply is regulated by the inflammatory milieu; the manuscript should address this interpretation.

3. Significance

Lines 414–467 (Discussion – Significance of findings):

The Discussion emphasizes the novel insight that OTUD7B simultaneously modulates inflammatory signaling (via RIPK1) and astrocytic structural stability (via GFAP). This dual role has far-reaching implications for CNS autoimmunity and potential therapeutic strategies in diseases such as multiple sclerosis.

Lines 493–500 (Additional discussion):

The authors emphasize the potential translational impact of their findings by demonstrating that OTUD7B influences key aspects of neuroinflammation, including enhanced chemokine signaling and increased immune cell recruitment. The reviewer notes that while the mechanistic insights are important, the manuscript does not address the therapeutic and translational relevance in depth. For instance, it remains unclear whether findings like regulation of STAT3 phosphorylation can be demonstrated in EAE tissue rather than just in vitro.

Reviewer's Comment:

The significance of the study would be enhanced by a discussion (or additional data) on the translational potential of these findings and by briefly contrasting these results with those on other deubiquitinating enzymes (e.g., A20, OTUB1).

4. Data and Methodology

Lines 538–704 (Materials and Methods):

Detailed protocols are provided for animal models, astrocyte isolation, histology, flow cytometry, and molecular analyses. These descriptions support reproducibility, although the reviewer suggests that additional detail on normalization and quality control (particularly for the RNA-seq and spatial transcriptomics described in lines 674–683) would be beneficial.

Lines 674–683 (Transcriptome analysis):

The authors outline the RNA-seq and spatial transcriptomics procedures. However, explicit information regarding normalization procedures, replicate numbers, and statistical quality control for these omics experiments should be added.

Reviewer's Comment:

Supplementary figures (e.g., Suppl. Fig.1) further support the data quality, but quantitative WB data (including proper quantification of band intensities) is missing throughout. Providing these quantifications would strengthen the conclusions drawn from the western blots.

5. Analytical Approach

Lines 237–245 (Gene ontology and pathway analyses):

The use of DAVID and ClueGO for KEGG and gene ontology enrichment is appropriate and helps contextualize the changes observed in the transcriptomic data.

Lines 295–300 and 360–370 (In vitro TNF dose-response and temporal studies):

The sequential analysis of TNF signaling, with careful attention to time points and different TNF concentrations, is well designed and reinforces the dynamic regulatory role of OTUD7B in astrocytes.

Reviewer's Comment:

More details regarding statistical corrections for multiple testing in the RNA-seq data would help clarify the analytical rigor of these experiments.

6. Suggested Improvements

Methodological Details (refer to Lines 674–683 and 225–234):

The manuscript would benefit from a clearer explanation of normalization procedures and the exact replicate numbers used in the RNA-seq and spatial transcriptomics analyses.

Streamlining Redundancies (Lines 414–467 vs. 493–500):

The Discussion contains some repetitive statements regarding the significance of the findings. Condensing these sections would improve clarity without sacrificing essential detail.

Additional points raised by the reviewer:

The title term “hyperinflammation” might be replaced with “enhanced inflammation” for clarity and consistency with common usage.

In lines 150/151 and Figure 1A, the manuscript refers to human tissue from an “anti-MOG mediated form of MS.” The reviewer requests clarification as to whether these patients represent typical MS cases or rather MOG-antibody associated disease (MOGAD), as these are distinct entities. The Methods section should be updated accordingly.

Lines 180/181 “suggesting that OTUD7B is a regulator of neuroinflammation” – how do the authors come to this conclusion? Why could it not be that OTUD7B is regulated by neuroinflammation, not a regulator?

Line 284 mentions “demyelination” but the presented data do not show myelin staining. Either perform myelin staining or revise the text to reflect the actual findings.

The spatial transcriptomics data might also be used to identify the cellular source of TNF in the lesion core—a point that would further contextualize the in vitro findings on TNF signaling.

7. Clarity and Context

Overall Manuscript (Lines 1–1100):

The manuscript is generally well written and the results are presented in a logical order. The introduction and discussion effectively place the findings within the broader context of astrocyte biology and neuroinflammation.

8. References

Lines 712–962 (References List):

The manuscript provides a comprehensive and appropriate list of references, including seminal works and recent studies relevant to astrocyte-mediated neuroinflammation and ubiquitin regulation.

Overall Recommendation

The manuscript offers significant and novel insights into the dual protective roles of astrocytic OTUD7B in CNS autoimmunity. The mechanistic study is well performed and the conclusions are supported by robust experimental evidence. However, to further strengthen the manuscript, I recommend major revisions to address the following points:

Enhance methodological transparency (replicate numbers, normalization procedures, and quantitative WB data).

Discuss the therapeutic and translational potential of the findings, possibly by demonstrating key *in vitro* findings (e.g., regulation of STAT3 phosphorylation) in EAE tissue.

Address minor textual and conceptual issues (clarification of patient classification regarding anti-MOG mediated disease vs. MOGAD, reconsidering the term “hyperinflammation,” and revising the description of demyelination if myelin staining is absent).

Finally, since the authors show the importance of TNF in the regulation of OTUD7B mediated pathways - What is the cellular source of TNF in the lesion core during EAE? Using their spatial transcriptomics studies might be helpful to determine major sources of TNF in the tissue.

Reviewer #2

(Remarks to the Author)

In the study “The deubiquitinase OTUD7B ameliorates central nervous system autoimmunity by inhibiting degradation of glial fibrillary acidic protein and astrocyte hyperinflammation”, Harit et al. investigated the impact of the deubiquitinase OTUD7B in astrocytes on neuroinflammation. They used a conditional gene knockout (GFAP-cre *Otud7b*^{fl/fl} mice), the experimental autoimmune encephalomyelitis mouse model, spatial transcriptomics, qRT-PCR, western blot techniques, flow cytometry and RNA-seq experiments of primary astrocytes. This manuscript explores a highly relevant topic and presents novel aspects that contribute valuable insights to the field. However, there are several critical points that require further clarification and revision before the paper can be considered for publication.

- When assessing the GSE32915 dataset description via <https://www.ncbi.nlm.nih.gov/geo/query/acc.cgi?acc=GPL4133&view=data> the probe ID 120 is labeled as PRRG2 (proline rich and Gla domain 2) rather than *Otud7b* (Fig. 1A)? Please clarify and discuss this discrepancy.
- The author should specify the technique used to determine the absolute cell count via flow cytometry in the Methods section (Fig. 4B/D).
- Why were only specific Western blots quantified and analyzed using appropriate statistical methods? Quantification and statistical analysis (n = 3) should be included more consistently rather than presenting only representative blots (e.g., Fig. 5B/C/D).
- I recommend rephrasing the sentences in lines 127-128: “In T cells, OTUD7B inhibits [?] T cell receptor (TCR) mediated activation by deubiquitination of ZAP70, a central molecule of proximal TCR signaling” and in lines 137-139: “Mechanistically OTUD7B inhibited proteasomal degradation of GFAP in reactive astrocytes by cleaving K48-polyubiquitin chains from GFAP and reducing GFAP mRNA production” to improve clarity and readability.
- The last part of the introduction (lines 134-145) reads more like a discussion rather than an introduction and should be revised accordingly.
- Typographical errors and inconsistencies:
 - o Should it be Suppl. Fig. 1D instead of 1E in line 267?
 - o Missing figure reference in line 282 (Fig. 4E?).
 - o Correction needed for OTUD7B in line 298.
 - o Should it be Suppl. Fig. 2A in line 331 instead of Suppl. Fig. 2B?
 - o Should it be Fig. 6F instead of Fig. 6E in line 360 and line 362 and line 365?
 - o Should it be Fig. 6G instead of Fig. 6F in line 372?
 - o There is no reference to Suppl. Fig. 2C.
 - o GFAP-cre *OTUD7b*^{fl/fl} mice in line 381?
 - o Figure Legend Figure 2A: the figure panel does not depict different spinal cord lesions compartments but rather different CNS-resident cell populations.
 - o Add a correct gamma in line 1044 should be edited.

Reviewer #3

(Remarks to the Author)

Reviewer #4

(Remarks to the Author)

The study explores the role of the deubiquitinating enzyme OTUD7B in astrocytes in EAE.

The main findings are that OTUD7B:

- is upregulated by astrocytes in inflammatory lesions of MS patients and EAE mice.
- limits neuroinflammation. In conditional KO of OTUD7B in astrocytes exacerbates EAE
- acts through 2 main mechanisms

Inhibition of TNF Signaling via suppressed TNF-induced chemokine production in astrocytes through sequential K63- and K48-deubiquitination of RIPK1, which reduces NF- κ B and MAPK activation.

Promotes expression of GFAP by supporting GFAP mRNA expression and preventing proteasomal degradation by cleaving K48-polyubiquitin chains from GFAP.

Key experiments are:

- RNA Expression Analysis of OTUD7B in MS and EAE using a publicly available microarray dataset of MOGAD patient and controls, and qRT-PCR and spatial transcriptomics (Xenium platform) in EAE tissue.
- Astrocyte-Specific OTUD7B Deletion in EAE, which results in a substantially higher clinical score, lower expression of GFAP in astrocytes, increased demyelination, larger inflammatory infiltrates and increased cytokine chemokine production. Ontology and KEGG pathway enrichment analysis demonstrated upregulation of pathways related to chemokine/cytokine signaling in OTUD7B-deficient astrocytes in EAE.
- In astrocyte cultures, OTUD7B suppressed TNF Signaling and NF- κ B and MAPK activation via by sequential K63 and K48 de-ubiquitination of RIPK1. Moreover, OTUD7B prevented GFAP mRNA and proteasomal degradation by K48 de-ubiquitination.

There are several major gaps that should be addressed:

- IHC should be performed for OTUD7B in EAE or MS to demonstrated expression on a protein level.
- What is the justification to examining datasets from MOGAD patient rather than MS?

How many MOGAD patients were examined in the published dataset?

Do more recent 10X snRNA-Seq data sets show similar OTUD7B upregulation?

- A major unanswered question is whether overexpression OTUD7B ameliorates EAE and/or TNF signaling in astrocyte culture.
- Spatial transcriptomics only provides so much insight into OTUD7B expression. The gold standard for demonstrating knock down of gene expression is genotyping of mice.
- It is unclear whether spatial transcriptomics was done in OTUD7B^{fl/fl} and KO mice. Line 975 in the figure legend states that it was done in both genotypes.
- The transcriptomic analysis of isolated astrocytes (Fig. 3) is very limited. At a minimum, it should include volcano plots of differential expression and GSEA profiling of the different groups. Single nucleus instead of bulk RNA- seq data would be preferable, especially if no spatial transcriptomics are available for KO mice.

Minor points:

- GFAP spelled wrong in Fig. 2F
- Annotation of Fig. 1 needs formatting
- Fig 1 A. Is OTUD7B mRNA expression (Y axis) in CPM?

Version 1:

Reviewer comments:

Reviewer #1

(Remarks to the Author)

My comments and concerns have been addressed sufficiently in the revised version of the manuscript. I have no further comments and recommend to accept the manuscript in its current form.

Reviewer #2

(Remarks to the Author)

All of my previously raised concerns have been adequately and thoughtfully addressed in the revised manuscript. The authors have incorporated the necessary revisions in a coherent and scientifically sound manner. I therefore consider the manuscript to be suitable for publication in its current form.

Reviewer #3

(Remarks to the Author)

Response to Referees comments

We thank you and the reviewers for critically evaluating our manuscript and providing valuable feedback. The reviewers raised important and specific concerns, which we have all carefully addressed. In response, we have performed additional experiments, incorporated new data, and provided the missing and supplementary information. We believe these modifications have significantly improved the quality and impact of the manuscript.

REVIEWER COMMENTS

Reviewer #1 (Remarks to the Author):

1. Key Results

Reviewer's Overview:

In their manuscript “The deubiquitinase OTUD7B ameliorates central nervous system autoimmunity by inhibiting degradation of glial fibrillary acidic protein and astrocyte hyperinflammation”, Harit and colleagues demonstrate that OTUD7B is upregulated in human and murine inflammatory CNS disorders. They show that deletion of OTUD7B in astrocytes (using GFAP-cre *Otud7b^{fl/fl}* mice) leads to more severe EAE with altered astrocyte morphology, reduced GFAP expression, and enhanced cytokine/chemokine production that in turn increases lymphocytic and monocytic recruitment into the CNS.

2. Validity

Lines 183–222 (Astrocyte-specific deletion and in vivo validation):

The generation and validation of the GFAP-cre *Otud7b^{fl/fl}* mouse model are clearly presented. The in vivo data support the claim that loss of OTUD7B in astrocytes exacerbates EAE pathology.

Lines 225–285 (Chemokine production and leukocyte infiltration):

Data showing increased chemokine mRNA levels and enhanced recruitment of encephalitogenic CD4⁺ T cells further support the authors' conclusions regarding the regulatory role of OTUD7B.

Lines 287–374 (TNF signaling analyses):

Detailed biochemical assays illustrate that OTUD7B regulates TNF-induced signaling by modulating the ubiquitination state of RIPK1. This mechanistic insight is critical for understanding the downstream effects on NF- κ B activation.

Reviewer's Comment:

Although the experimental design is robust overall, additional clarity on replicate numbers and data normalization—especially for the RNA-seq and spatial transcriptomics experiments—is needed to further reinforce the data's robustness.

Response: We have revised the methods section and included the number of replicates and the data normalization protocols. (both in M&M, **page 31**, & **page 32**)

For RNA-seq, astrocytes were isolated by magnetic microbeads from spinal cords of *Otud7b^{fl/fl}* and GFAP-cre *Otud7b^{fl/fl}* mice, respectively, at day 15 p.i. Astrocytes isolated from non-immunized *Otud7b^{fl/fl}* and GFAP-cre *Otud7b^{fl/fl}* mice were used as controls. For each experimental group, astrocytes from three mice (n=3) were pooled to reduce biological variability, resulting in one sample per condition. These information

are now included on **page 30, line 697-702**. Pooling small RNA samples is effective in reducing sample variability and compensates for the requirement of high replicate number (<https://doi.org/10.1186/s12864-020-6721-y>).

RNA-seq data were normalized based on log₂ counts per million (CPM) values and quantile normalization (edgeR). Based on the 20th percentile of distribution of the normalized log₂ CPM values a CPM cutoff of 0.56 was derived to discriminate low abundant transcripts. Only transcripts with log₂ CPM > 0.56 in were used for further analysis. Differential expression between samples from GFAP-cre Otud7b^{fl/fl} versus Otud7b^{fl/fl} control mice isolated on day 0 and day 15 was calculated, respectively. Transcripts with fold change (FC) > 3 were regarded relevant.

This information is now included on **page 31, lines 716 to 723**.

To confirm and extend the RNA-seq data, we decided to perform spatial transcriptomics to delineate the regional distribution of OTUD7b-regulated astrocyte reactions in EAE. Spatial transcriptomics was performed on spinal cords of Otud7b^{fl/fl} and GFAP-cre Otud7b^{fl/fl} mice (n=3), respectively, at day 15 p.i. using the Xenium system (10x Genomics). Astrocytes isolated from non-immunized Otud7b^{fl/fl} and GFAP-cre Otud7b^{fl/fl} mice were used as control (n=3). On **page 31, lines 728 to 731**.

Spatial data were processed using the VoltRon package. Differential expression was calculated using DESeq2 based on "pseudobulk" raw counts (i.e., per cell type and per sample, counts are added up for all cells). Count normalization per gene was done using the formula $\log((\text{raw count for the gene})/(\text{raw count sum for the cell}) * (\text{size factor}) + 1)$, with a size factor of 10.000, and using natural logarithms. This information is now included on **page 32 lines 742-748**.

Moreover, the reviewer questions whether OTUD7B is truly acting as a regulator of neuroinflammation (as claimed in lines 180/181) or simply is regulated by the inflammatory milieu; the manuscript should address this interpretation.

Response: While OTUD7B is upregulated in multiple sclerosis and experimental autoimmune encephalomyelitis (EAE), its active regulatory role in astrocytes is functionally supported by our in vitro data, which show that OTUD7B significantly reduces TNF-induced NF- κ B activation and chemokine production (Fig. 7A, B) and GFAP degradation (Fig. 8B,C). These findings demonstrate that OTUD7B is not only regulated by the inflammatory milieu but also acts as a critical regulator of neuroinflammation.

We agree with the reviewer that, our statement on line 181-183 (previous version) is too early positioned in the manuscript to conclude that OTUD7B is a regulator of neuroinflammation. We have revised this section and it now reads: "Taken together, astrocytes upregulated Otud7b expression in CNS autoimmunity with the highest expression in the T cell-enriched lesions and a gradual decline with increasing distance from the lesion cores." (**Page 8, lines 173-175** of the revised manuscript.)

3. Significance

Lines 414–467 (Discussion – Significance of findings):

The Discussion emphasizes the novel insight that OTUD7B simultaneously modulates inflammatory signaling (via RIPK1) and astrocytic structural stability (via GFAP). This dual role has far-reaching implications for CNS autoimmunity and potential therapeutic strategies in diseases such as multiple sclerosis.

Lines 493–500 (Additional discussion):

The authors emphasize the potential translational impact of their findings by demonstrating that OTUD7B influences key aspects of neuroinflammation, including enhanced chemokine signaling and increased immune cell recruitment.

The reviewer notes that while the mechanistic insights are important, the manuscript does not address the therapeutic and translational relevance in depth. For instance, it remains unclear whether findings like regulation of STAT3 phosphorylation can be demonstrated in EAE tissue rather than just in vitro.

Response: According to the reviewer's suggestion to address the therapeutic and translational relevance of our findings in vivo, we performed additional immunohistochemistry (IHC) analyses of spinal cord tissue and assessed the expression of OTUD7B in astrocytes, and its impact on phosphorylation of STAT3 (pSTAT3). Consistent with our RNA data, OTUD7B protein levels were upregulated in astrocytes of Otud7b-competent mice during EAE (new Fig. 1C). In addition, our new in vivo immunofluorescence data demonstrate increased expression of OTUD7b and GFAP in Otud7b^{fl/fl} as compared to GFAP-cre Otud7b^{fl/fl} mice at day 15 of EAE (new Fig.8D). These in vivo data confirm our in vitro analyses, that OTUD7b positively regulates STAT3 phosphorylation and GFAP expression upon prolonged TNF exposure (Fig. 8B).

Reviewer's Comment:

The significance of the study would be enhanced by a discussion (or additional data) on the translational potential of these findings and by briefly contrasting these results with those on other deubiquitinating enzymes (e.g., A20, OTUB1).

Response: We have expanded the discussion on **page 23, lines 530-540** to elaborate on therapeutic implications, such as targeting OTUD7B to modulate astrocyte reactivity in MS, and compare our findings with other deubiquitinases including A20 and OTUB1 to highlight OTUD7B's unique dual regulatory impact on inflammatory and structural astrocyte functions by sequential K63- and K48-deubiquitination of RIPK1 and by directly stabilizing GFAP protein through K48-deubiquitination

4. Data and Methodology

Lines 538–704 (Materials and Methods):

Detailed protocols are provided for animal models, astrocyte isolation, histology, flow cytometry, and molecular analyses. These descriptions support reproducibility, although the reviewer suggests that additional detail on normalization and quality control (particularly for the RNA-seq and spatial transcriptomics described in lines 674–683) would be beneficial.

Lines 674–683 (Transcriptome analysis):

The authors outline the RNA-seq and spatial transcriptomics procedures. However, explicit information regarding normalization procedures, replicate numbers, and statistical quality control for these omics experiments should be added.

Response: The details of normalization and quality control are now included in the methods section. In addition validity, normalization and quality control are now described in more detail **on page 31 and 32**, for RNA-seq and **on page 31, lines 716 to 723**, and **page 32 lines 742-747** for spatial transcriptomics.

Reviewer's Comment:

Supplementary figures (e.g., Suppl. Fig.1) further support the data quality, but quantitative WB data (including proper quantification of band intensities) is missing throughout. Providing these quantifications would strengthen the conclusions drawn from the western blots.

Response: We have included triplicates for every single western blot; the additional two WB are provided in the supplementary figures. All WB have been quantified accordingly. In summary, all of the WB support our conclusions drawn from the WB shown before.

5. Analytical Approach

Lines 237–245 (Gene ontology and pathway analyses):

The use of DAVID and ClueGO for KEGG and gene ontology enrichment is appropriate and helps contextualize the changes observed in the transcriptomic data.

Lines 295–300 and 360–370 (In vitro TNF dose-response and temporal studies):

The sequential analysis of TNF signaling, with careful attention to time points and different TNF concentrations, is well designed and reinforces the dynamic regulatory role of OTUD7B in astrocytes.

Reviewer’s Comment:

More details regarding statistical corrections for multiple testing in the RNA-seq data would help clarify the analytical rigor of these experiments.

Response: Only genes showing counts greater than 5 were considered for further analysis. Gene annotation was done by R package bioMaRt. Data normalization was performed by calculating log₂ counts per million (CPM) values and quantile normalization (edgeR). Based on the 20th percentile of distribution of the normalized log₂ CPM values a CPM cutoff of 0.56 was derived to discriminate lowly abundant transcripts. Only transcripts with log₂ CPM > 0.56 in at least one sample were used for further analysis. Differential expression between samples from Otud7b knockout versus Otud7b-competent mice isolated on day 0 and day 15 was calculated, respectively. Transcripts with fold change (FC) with $|FC| > 3$ were regarded relevant. This information is now included in the methods section on **page 31 Lines 714 to 723**.

6. Suggested Improvements

Methodological Details (refer to Lines 674–683 and 225–234):

The manuscript would benefit from a clearer explanation of normalization procedures and the exact replicate numbers used in the RNA-seq and spatial transcriptomics analyses.

The details of normalization and quality control are now included in the methods section. In addition validity, normalization and quality control are now described in more detail on **page 30 and 31 lines 698-725** for RNA-seq and on **pages 31 and 32, lines 727-7476**, for spatial transcriptomics.

Streamlining Redundancies (Lines 414–467 vs. 493–500):

The Discussion contains some repetitive statements regarding the significance of the findings. Condensing these sections would improve clarity without sacrificing essential detail.

Response: We have removed the repetitive statements on lines 478-471,472-477, 501-503 and condensed these sections.

Additional points raised by the reviewer:

The title term “hyperinflammation” might be replaced with “enhanced inflammation” for clarity and consistency with common usage.

Response: The word “hyperinflammation” has been changed to “enhanced inflammation” in the **title, on page 45, lines 1180 and in Figure 9.**

In lines 150/151 and Figure 1A, the manuscript refers to human tissue from an “anti-MOG mediated form of MS.” The reviewer requests clarification as to whether these patients represent typical MS cases or rather MOG-antibody associated disease (MOGAD), as these are distinct entities. The Methods section should be updated accordingly.

Response: We agree with the reviewer that the data of MOGAD patient presented in Fig. 1A of the original manuscript are not representative for a typical form of MS. Since the EAE model used in our study most closely resembles progressive MS, we have replaced Figure 1A and now include a new Figure 1A, which shows single-cell transcriptomic data from patients with progressive MS. Our analyses revealed enhanced expression of *Otud7b* of astrocytes in active MS lesions and in the periplaque white matter as compared to healthy individuals.

Lines 180/181 “suggesting that OTUD7B is a regulator of neuroinflammation” – how do the authors come to this conclusion? Why could it not be that OTUD7B is regulated by neuroinflammation, not a regulator?

Response: We agree with the reviewer that, at this point in the manuscript, it can only be concluded that in EAE *Otud7b* is upregulated in astrocytes but that it is too early to state at this point of the manuscript that OTUD7B is a regulator of neuroinflammation. We have revised this section accordingly. It now reads: “Taken together, astrocytes upregulate *Otud7b* expression in CNS autoimmunity with the highest expression in the T cell-enriched lesions and a gradual decline with increasing distance from the lesions core.” (**Page 8, lines 173-175.**)

Line 284 mentions “demyelination” but the presented data do not show myelin staining. Either perform myelin staining or revise the text to reflect the actual findings.

Response: Luxol fast blue stains myelin. Since the thickness of myelin layer is reduced in GFAP-cre *Otud7b^{fl/fl}* mice, we replaced “demyelination” by “reduced myelin thickness” **on page 9, line 207.**

The spatial transcriptomics data might also be used to identify the cellular source of TNF in the lesion core—a point that would further contextualize the in vitro findings on TNF signaling.

Response: As suggested, we have reanalyzed the spatial transcriptomic data and identified microglia/macrophages and T cells as the major cellular sources of TNF (new Fig. 7A). The percentage of TNF-producing cells was similar in both *Otud7b^{fl/fl}* and GFAP-cre *Otud7b^{fl/fl}* mice. However, the increased infiltration of microglia/macrophages and $CD3^+$ T cells in the knockout mice upon EAE (Fig. 4B) resulted in higher overall TNF levels at day 15 p.i. (Fig. 7B).

7. Clarity and Context

Overall Manuscript (Lines 1–1100):

The manuscript is generally well written and the results are presented in a logical order. The introduction and discussion effectively place the findings within the broader context of astrocyte biology and neuroinflammation.

8. References

Lines 712–962 (References List):

The manuscript provides a comprehensive and appropriate list of references, including seminal works and recent studies relevant to astrocyte-mediated neuroinflammation and ubiquitin regulation.

Overall Recommendation

The manuscript offers significant and novel insights into the dual protective roles of astrocytic OTUD7B in CNS autoimmunity. The mechanistic study is well performed and the conclusions are supported by robust experimental evidence. However, to further strengthen the manuscript, I recommend major revisions to address the following points:

Enhance methodological transparency (replicate numbers, normalization procedures, and quantitative WB data).

Discuss the therapeutic and translational potential of the findings, possibly by demonstrating key *in vitro* findings (e.g., regulation of STAT3 phosphorylation) in EAE tissue.

Address minor textual and conceptual issues (clarification of patient classification regarding anti-MOG mediated disease vs. MOGAD, reconsidering the term “hyperinflammation,” and revising the description of demyelination if myelin staining is absent).

Finally, since the authors show the importance of TNF in the regulation of OTUD7B mediated pathways - What is the cellular source of TNF in the lesion core during EAE? Using their spatial transcriptomics studies might be helpful to determine major sources of TNF in the tissue.

Response: We have implemented all major revisions as suggested and discussed above, including enhanced methodological transparency, expanded translational relevance, and textual clarifications to strengthen the manuscript. The details of these changes are provided in the respective sections above.

Reviewer #2 (Remarks to the Author):

In the study “The deubiquitinase OTUD7B ameliorates central nervous system autoimmunity by inhibiting degradation of glial fibrillary acidic protein and astrocyte hyperinflammation”, Harit et al. investigated the impact of the deubiquitinase OTUD7B in astrocytes on neuroinflammation. They used a conditional gene knockout (GFAP-cre *Otud7b*^{fl/fl} mice), the experimental autoimmune encephalomyelitis mouse model, spatial transcriptomics, qRT-PCR, western blot techniques, flow cytometry and RNA-seq experiments of primary astrocytes. This manuscript explores a highly relevant topic and presents novel aspects that contribute valuable insights to the field. However, there are several critical points that require further clarification and revision before the paper can be considered for publication.

- When assessing the GSE32915 dataset description via <https://www.ncbi.nlm.nih.gov/geo/query/acc.cgi?acc=GPL4133&view=data> the probe ID 120 is labeled as PRRG2 (proline rich and Gla domain 2) rather than *Otud7b* (Fig. 1A)? Please clarify and discuss this discrepancy.

Response: We thank the reviewer for pointing this out. We have rectified our mistake and now include a new with Figure 1A, which shows single-cell transcriptomic data from patients with progressive MS (GSE180759) in which we detected enhanced expression of OTUD7B in astrocytes.

- The author should specify the technique used to determine the absolute cell count via flow cytometry in the Methods section (Fig. 4B/D).

Response: Cells were isolated from the spinal cords by mincing tissue through cell strainer followed by Percoll gradient centrifugation and counting of trypan blue-stained cells microscopically in a

hemocytometer. We calculate absolute numbers of individual cell populations based on the percentage of the cell population in flow cytometry and the absolute cell number determined microscopically. This method has been standardized in our lab and results in minimal cell loss (<https://doi.org/10.15252/embj.2018100947> DOI: 10.1007/s00401-013-1183-9). This information is included on Page21, lines 616-631.

- Why were only specific Western blots quantified and analyzed using appropriate statistical methods? Quantification and statistical analysis (n = 3) should be included more consistently rather than presenting only representative blots (e.g., Fig. 5B/C/D).

Response: We have included triplicates for each single WB; the additional two WBs are provided in the supplementary figures. All WBs have been quantified accordingly.

- I recommend rephrasing the sentences in lines 127-128: “In T cells, OTUD7B inhibits [?] T cell receptor (TCR) mediated activation by deubiquitination of ZAP70, a central molecule of proximal TCR signaling”

Response: We have revised the sentence to: “In T cells, OTUD7B inhibits T cell receptor (TCR)- and CD28-mediated activation by regulating the non-degradative ubiquitination of ZAP70, a central molecule in proximal TCR signaling”, (**page 6, lines 127-132**).

and in lines 137-139: “Mechanistically OTUD7B inhibited proteasomal degradation of GFAP in reactive astrocytes by cleaving K48-polyubiquitin chains from GFAP and reducing GFAP mRNA production” to improve clarity and readability.

Response: As suggested by the reviewer, we have shortened this section and limited the mechanistic details to the discussion. Therefore, we have removed this sentence.

- The last part of the introduction (lines 134-145) reads more like a discussion rather than an introduction and should be revised accordingly.

Response: As per the reviewer’s suggestion, we have shortened this section and limited the mechanistic details to the discussion.

- Typographical errors and inconsistencies:

- o Should it be Suppl. Fig. 1D instead of 1E in line 267? Yes, has been corrected

- o Missing figure reference in line 282 (Fig. 4E?). Figure reference included.

- o Correction needed for OTUD7B in line 298. Has been corrected

- o Should it be Suppl. Fig. 2A in line 331 instead of Suppl. Fig. 2B? It has been correct according to the new supplementary figure 3C

- o Should it be Fig. 6F instead of Fig. 6E in line 360 and line 362 and line 365? Has been corrected

- o Should it be Fig. 6G instead of Fig. 6F in line 372? Has been corrected

- o There is no reference to Suppl. Fig. 2C. Has been corrected

- o GFAP-cre OTUD7bfl/fl mice in line 381? Has been corrected

- o Figure Legend Figure 2A: the figure panel does not depict different spinal cord lesions compartments but rather different CNS-resident cell populations. Has been corrected

- o Add a correct gamma in line 1044 should be edited ? Has been corrected

Reviewer #3 (Remarks to the Author):

Reviewer #4 (Remarks to the Author):

The study explores the role of the deubiquitinating enzyme OTUD7B in astrocytes in EAE.

The main findings are that OTUD7B:

- is upregulated by astrocytes in inflammatory lesions of MS patients and EAE mice.
- limits neuroinflammation. In conditional KO of OTUD7B in astrocytes exacerbates EAE
- acts through 2 main mechanisms

Inhibition of TNF Signaling via suppressed TNF-induced chemokine production in astrocytes through sequential K63- and K48-deubiquitination of RIPK1, which reduces NF- κ B and MAPK activation.

Promotes expression of GFAP by supporting GFAP mRNA expression and preventing proteasomal degradation by cleaving K48-polyubiquitin chains from GFAP.

Key experiments are:

- RNA Expression Analysis of OTUD7B in MS and EAE using a publicly available microarray dataset of MOGAD patient and controls, and qRT-PCR and spatial transcriptomics (Xenium platform) in EAE tissue.
- Astrocyte-Specific OTUD7B Deletion in EAE, which results in a substantially higher clinical score, lower expression of GFAP in astrocytes, increased demyelination, larger inflammatory infiltrates and increased cytokine chemokine production. Ontology and KEGG pathway enrichment analysis demonstrated upregulation of pathways related to chemokine/cytokine signaling in OTUD7B-deficient astrocytes in EAE.
- In astrocyte cultures, OTUD7B suppressed TNF Signaling and NF- κ B and MAPK activation via by sequential K63 and K48 de-ubiquitination of RIPK1. Moreover, OTUD7B prevented GFAP mRNA and proteasomal degradation by K48 de-ubiquitination.

There are several major gaps that should be addressed:

- IHC should be performed for OTUD7B in EAE or MS to demonstrated expression on a protein level.

Response: According to the reviewer's suggestion, we performed additional immunohistochemistry (IHC) analyses on spinal cord tissue from EAE mice to assess the expression of OTUD7B. Consistent with our RNA data, OTUD7B protein levels were upregulated in astrocytes during EAE. The data is now displayed in new Fig. 1C.

- What is the justification to examining datasets from MOGAD patient rather than MS?

How many MOGAD patients were examined in the published dataset?

Do more recent 10X snRNA-Seq data sets show similar OTUD7B upregulation?

Response: We agree with the reviewer, that the selection of a MOGAD data set was not appropriate. Therefore, we include a new Figure 1A, which shows single-cell transcriptomic data of astrocytes from patients with progressive MS. Our analysis shows enhanced expression of OTUD7B in astrocytes in MS compared to white matter astrocytes of healthy controls.

- A major unanswered question is whether overexpression OTUD7B ameliorates EAE and/or TNF signaling in astrocyte culture.

Response: As suggested, we overexpressed OTUD7B in cultivated OTUD7b-competent and -deficient astrocytes and stimulated the cells with TNF. Our new data show that OTUD7B overexpression suppressed TNF-induced phosphorylation of p65 in both OTUD7B competent- and -deficient astrocytes (Fig. 6A). In addition, OTUD7b overexpression suppressed TNF-induced chemokine (Cxcl1, Cxcl10, Cxcl11, Ccl2,

Ccl20), NOS2 and IL-6 mRNA production in both genotypes (Fig. 6B). This provides additional functional evidence that OTUD7B negatively regulates TNF signaling.

- Spatial transcriptomics only provides so much insight into OTUD7B expression. The gold standard for demonstrating knock down of gene expression is genotyping of mice.

Response: We agree with the reviewer's comment. All mice used in the present study were genotyped by PCR as described in Material and Methods (**page 24 and 25**). We added the primers used for genotyping of mice to Material and Methods (**page 24 and 25, lines 563-568**) and show it in Suppl. Fig. 1A results of the genotyping PCR. Additionally, the absence of OTUD7B protein in cultivated astrocytes of GFAP-cre Otud7b^{fl/fl} astrocytes is shown in Supplementary Figure 1B (WB), Suppl. Fig. 1C (IHC), Fig. 6A, Fig. 7C, 7E, 7G, Fig. 8B (all WB).

- It is unclear whether spatial transcriptomics was done in OTUD7Bfl/fl and KO mice. Line 975 in the figure legend states that it was done in both genotypes.

Response: The spatial transcriptome data presented in Figure 1 were obtained from Otud7b^{fl/fl} mice, whereas the data in Figures 2, 3, 6, and 7 were derived from both Otud7b^{fl/fl} and GFAP-cre Otud7b^{fl/fl} mice.

- The transcriptomic analysis of isolated astrocytes (Fig. 3) is very limited. At a minimum, it should include volcano plots of differential expression and GSEA profiling of the different groups. Single nucleus instead of bulk RNA-seq data would be preferable, especially if no spatial transcriptomics are available for KO mice.

Response: Volcano plots could not be generated because the RNA-seq data were derived from pooled samples (n=3). Single-nucleus RNA-seq was not conducted; however, spatial transcriptomics data from knockout (KO) mice, as shown in Figures 2, 3, 6, and 7, provide valuable spatial context for gene expression analysis

Minor points:

- GFAP spelled wrong in Fig. 2F. Has been corrected
- Annotation of Fig. 1 needs formatting. Has been corrected
- Fig 1 A. Is OTUD7B mRNA expression (Y axis) in CPM? Yes. However as suggested by the reviewer's, we added a new Fig. 1A showing OTUD7B mRNA expression in progressive MS patients.